# Defense against phytopathogens relies on efficient antimicrobial protein secretion mediated by the microtubule-binding protein TGNap1

Deepak D. Bhandari [1,2], Dae Kwan Ko [1,2], Sang-Jin Kim [1,2,3], Kinya Nomura[4,5], Sheng Yang He [4,5] & Federica Brandizzi [1,2,3] ✉

Plant immunity depends on the secretion of antimicrobial proteins, which occurs through yet-largely unknown mechanisms. The *trans*-Golgi network (TGN), a hub for intracellular and extracellular trafficking pathways, and the cytoskeleton, which is required for antimicrobial protein secretion, are emerging as pathogen targets to dampen plant immunity. In this work, we demonstrate that *tgnap1-2*, a loss-of-function mutant of *Arabidopsis* TGNap1, a TGN-associated and microtubule (MT)-binding protein, is susceptible to *Pseudomonas syringae* (*Pst* DC3000). *Pst* DC3000 infected *tgnap1-2* is capable of mobilizing defense pathways, accumulating salicylic acid (SA), and expressing antimicrobial proteins. The susceptibility of *tgnap1-2* is due to a failure to efficiently transport antimicrobial proteins to the apoplast in a partially MT-dependent pathway but independent from SA and is additive to the pathogen-antagonizing MIN7, a TGN-associated ARF-GEF protein. Therefore, our data demonstrate that plant immunity relies on TGNap1 for secretion of antimicrobial proteins, and that TGNap1 is a key immunity element that functionally links secretion and cytoskeleton in SA-independent pathogen responses.

Biotic diseases lead to major crop losses worldwide and thus reduce food production. Therefore, understanding the fundamental mechanisms underpinning plant defense is critical to designing sustainable crop management systems. Over the course of evolution, plant-pathogen interactions have shaped the plant immune system against diverse pathogens and led to the development of bacterial strategies to suppress plant immunity. Activation of plant immunity is surveilled by surface and intracellular receptors[1–3]. Recognition of conserved microbial patterns (e.g., flagellin, chitin) by plasma membrane (PM) receptors leads to the activation of pattern-triggered immunity (PTI), which activates signaling cascades to limit pathogen growth. To suppress PTI, pathogens secrete effectors into host cells to target organelles and signaling nodes, dampen PTI, and enhance pathogen virulence[4,5]. In turn, pathogen effectors are recognized by the host's nucleotide-binding leucine-rich repeat (NLR) proteins resulting in an activation of effector-triggered immunity (ETI). Activation of ETI amplifies defense activation by a rapid activation of signaling cascades, re-routing intracellular traffic, and, in some cases, localized cell death[6–8]. A state of activated defense that limits infection of virulent pathogen is defined as basal immunity. Significant advances have been made in identifying proteins and signaling cascades involved in PTI and ETI, and the

[1]MSU-DOE Plant Research Laboratory, Michigan State University, East Lansing, MI 48824, USA. [2]Great Lakes Bioenergy Research Center, Michigan State University, East Lansing, MI 48824, USA. [3]Department of Plant Biology, Michigan State University, East Lansing, MI 48824, USA. [4]Department of Biology, Duke University, Durham, NC 27708, USA. [5]Howard Hughes Medical Institute, Duke University, Durham, NC 27708, USA. ✉e-mail: fb@msu.edu

importance of intracellular trafficking as a pathogen target is being recognized[8–10].

During immunity, antimicrobial proteins and proteases are secreted to the apoplast, which is a niche for non-penetrating bacterial pathogens[11–13]. Despite the essential role of the plant secretory pathway in the trafficking of antimicrobial proteins, significant gaps in the understanding of the mechanisms underlying protein traffic in defense from bacteria still exist. Conventional protein secretion begins at the endoplasmic reticulum (ER); proteins are then trafficked to the Golgi prior to sorting at the *trans*-Golgi network (TGN) and secretory vesicles for delivery to the cell surface through yet-largely uncharacterized mechanisms[14]. In plant cells, post-Golgi protein secretion depends on the cytoskeleton components actin and microtubules (MTs)[15,16], and chemical disruption of actin or MTs increases susceptibility to pathogens[17–20], possibly due to a disruption of membrane traffic. Cytoskeleton-mediated traffic requires accessory proteins that connect the cytoskeleton with vesicles. Nonetheless, the identity of cytoskeleton-interacting proteins required for efficient traffic of antimicrobial proteins is still largely unknown. The plant TGN encompasses subpopulations of vesicles that are only partially overlapping[15,21]. Whether the various TGN subpopulations have overlapping roles in plant immunity is still an open question. Also, only a few TGN proteins with roles inactivation or repression of immune signaling have been identified to date. For instance, a loss of the TGN protein ECHIDNA results in a constitutive accumulation of the phytohormone salicylic acid (SA) and cell death, leading to constitutively activated defense[22]. In contrast, the TGN/early endosome (EE) localized ARF-GEF (guanine-nucleotide exchange factors for ADP-ribosylation factor GTPases) protein HOPM1-INTERACTOR 7 (MIN7) is involved in both PTI and ETI[23]. The loss of MIN7 does not completely abrogate immunity, supporting that yet-unknown proteins operate at the TGN to facilitate the secretion of antimicrobial proteins.

To contribute to the understanding of the mechanisms underlying the traffic of antimicrobial proteins during plant immunity, we characterized the role of TGNap1 in plant immunity. TGNap1 is a TGN-associated protein that interacts with MTs and is key for the biogenesis of a subset of TGNs[15]. Here we report that TGNap1 plays a critical role in immunity, which is independent of SA and additive to MIN7, wherein TGNap1 is required for MT-mediated secretion of antimicrobial proteins.

## Results

### TGNap1 is required for a robust immune response
In an earlier study, TGNap1, a TGN-associated protein, has been shown to bind MTs and is required for the homeostasis of the endocytic and exocytic traffic[15]. Therefore, we adopted TGNap1 as a tool to gain insights into TGN-dependent mechanisms for the traffic of antimicrobial proteins in plant immunity. To do so, we infected a complete *TGNap1* knockout (*tgnap1-2*) with the virulent pathogen *Pst* DC3000. Compared to Col-3 (wild-type; WT), we found that *tgnap1-2* was more susceptible to infection by *Pst* DC3000 than WT, displaying eight–tenfold increased bacterial growth but less than the hypersusceptible mutant *enhanced disease susceptibility 1* (*eds1-2*)[24] (Fig. 1a). To test if differences ecotypes could account for the increased bacterial growth, we infected Col-0 and Col-3 plants with *Pst* DC3000 and observed that both Col-0 and Col-3 plants had comparable susceptibility, while *tgnap1-2* and *eds1-2* (in Col-0) were hypersusceptible (Supplementary Fig. S1a). Complementing *tgnap1-2* with a full-length TGNap1 fused to YFP (TGNap1-YFP) restored the susceptibility of *tgnap1-2* to WT levels (Fig. 1a and Supplementary Fig. S1b). Thus, TGNap1 is a critical component of plant immunity.

Because we found that *tgnap1-2* is hypersusceptible to *Pst* DC3000 (Fig. 1a and Supplementary Fig. S1), we next tested the involvement of TGNap1 in ETI and PTI. To do so, we infiltrated WT and *tgnap1-2* plants with *Pst* AvrRps4 and *Pst* AvrRpt2, which are

recognized by the products of Toll-Interleukin-1 receptor domain-NLR gene pair, *RPS4* (Resistance to *Pseudomonas syringae* 4) and *RRS1*(resistance to *Ralstonia solanacearum* 1)[25], and of the coiled-coil NLR *RPS2* (Resistance to *Pseudomonas syringae* 2)[26], respectively (Fig. 1b). Both *tgnap1-2* and TGNap1-YFP displayed WT-like resistance, while *eds1-2* and the SA-deficient mutant, *sid2-2*[27], were susceptible against *Pst* AvrRps4 (TNL) and *Pst* AvrRpt2 (CNL), respectively (Fig. 1b). These results indicate that, although *tgnap1-2* has a significant role in basal immunity against *Pst* DC3000, it is dispensable during ETI. Next, to assess a role of TGNap1 in PTI, we used flg22, a 22-amino acid peptide of flagellin, which activates PTI[28]. We infiltrated flg22 into Col, *tgnap1-2*, TGNap1-YFP, and *fec* (a triple mutant of PRR receptors *fls2, efr*, and *cerk1* (FLS2 – Flagellin sensing 2; EFR – Elongation factor-Tu receptor; CERK1- Chitin elicitor receptor kinase 1)[29]. Recognition of flg22 by FLS2 leads to rapid activation of mitogen-activated proteins kinase (MPK) signaling cascades[30]. Upon flg22 treatment, MPK3/MPK6 phosphorylation in *tgnap1-2* was indistinguishable from Col and TGNap1-YFP (Supplementary Fig. S1c), indicating that PTI-activated MPK signaling cascades are not affected by the loss of TGNap1. In addition, we tested if *tgnap1-2* was required in resistance to PTI-activating *Pst hrcC*, a *Pst* strain with a defective Type-III-secretion system[31]. Upon infection with *Pst hrcC*, the bacterial growth in *tgnap1-2* was comparable to Col-3 and TGNap1-YFP (Fig. 1c). The observed WT-like levels of bacterial growth of *tgnap1-2* to *Pst hrcC* is suggestive of one or more *Pst*-elicited effectors targeting TGNap1 functions or of the possibility that TGNap1 is recruited specifically in basal defense against *Pst* DC3000. To test if the immune role of TGNap1 was limited to *Pst* DC3000, we checked if *tgnap1-2* was susceptible to the Oomycete *Hyaloperonospora arabidopsis (Hpa)* NOCO2, which causes downy mildew[32]. Upon infection with *Hpa* NOCO2, the *tgnap1-2* mutant showed intermediate susceptibility between Col-3 and the hypersusceptible *eds1-2*, while no spores were observed in the *Hpa* NOCO2-resistant Ws-2 plants (Supplementary Fig. S1d). Collectively, these results indicate a functional role of TGNap1 in limiting virulent *Pst* DC3000 and *Hpa*, but not in ETI or *Pst hrcC*−mediated PTI.

Next, we aimed to gain insights on whether the observed hypersusceptibility of *tgnap1-2* to *Pst* DC3000 could be due to a specific pathogen-elicited effector. *Pst* DC3000 contains 36 effectors, of which 8 are sufficient to restore much of the *Pst* virulence in *Nicotiana benthamiana*[5,33,34]. We infected Col-3 and *tgnap1-2* plants with the following *Pst* mutants: *Pst avrPto avrPtoB* (encoding effectors involved in immune suppression)[35], *Pst avrE hopM1* (ΔEM) (encoding effectors involved in water-soaking and suppression of PTI)[36], and *Pst* ΔCEL (a deletion mutant of the conserved effector locus, including *avrE* and *hopM1*, leading to reduced virulence)[5]. Compared to *Pst* DC3000, both *Pst avrPto avrPtoB* and *Pst* ΔEM grew less on Col-3, underscoring their roles in virulence of *Pst* DC3000 in this ecotype (Supplementary Fig. S2 and Fig. 1a). However, *tgnap1-2* showed increased susceptibility, with 5-6 fold higher bacterial growth compared to Col-3 against both *Pst avrPto avrPtoB* and *Pst* ΔEM (Supplementary Fig. S2). Against *Pst* ΔCEL, *tgnap1-2* showed a significant but less pronounced increase in bacterial load compared to Col-3, supporting that, with reduced virulence, the pathogen growth differences between Col-3 and *tgnap1-2* were also reduced, but not absent. TGNap1 transcript and protein levels were unaltered upon *Pst* DC3000 infiltration (Fig. 1d and Supplementary Fig. S3). Taken together, these results underscore a critical role of TGNap1 in basal immunity, which is likely separate from canonical ETI and PTI defense pathways targeted by *Pst* effectors.

### Defects in immune signaling due to the loss of *TGNap1* are independent of SA and not alleviated upon induction of resistance
Because SA signaling pathways play a key role in limiting *Pst* DC3000 growth[27], we tested if *tgnap1-2* was defective in activating or

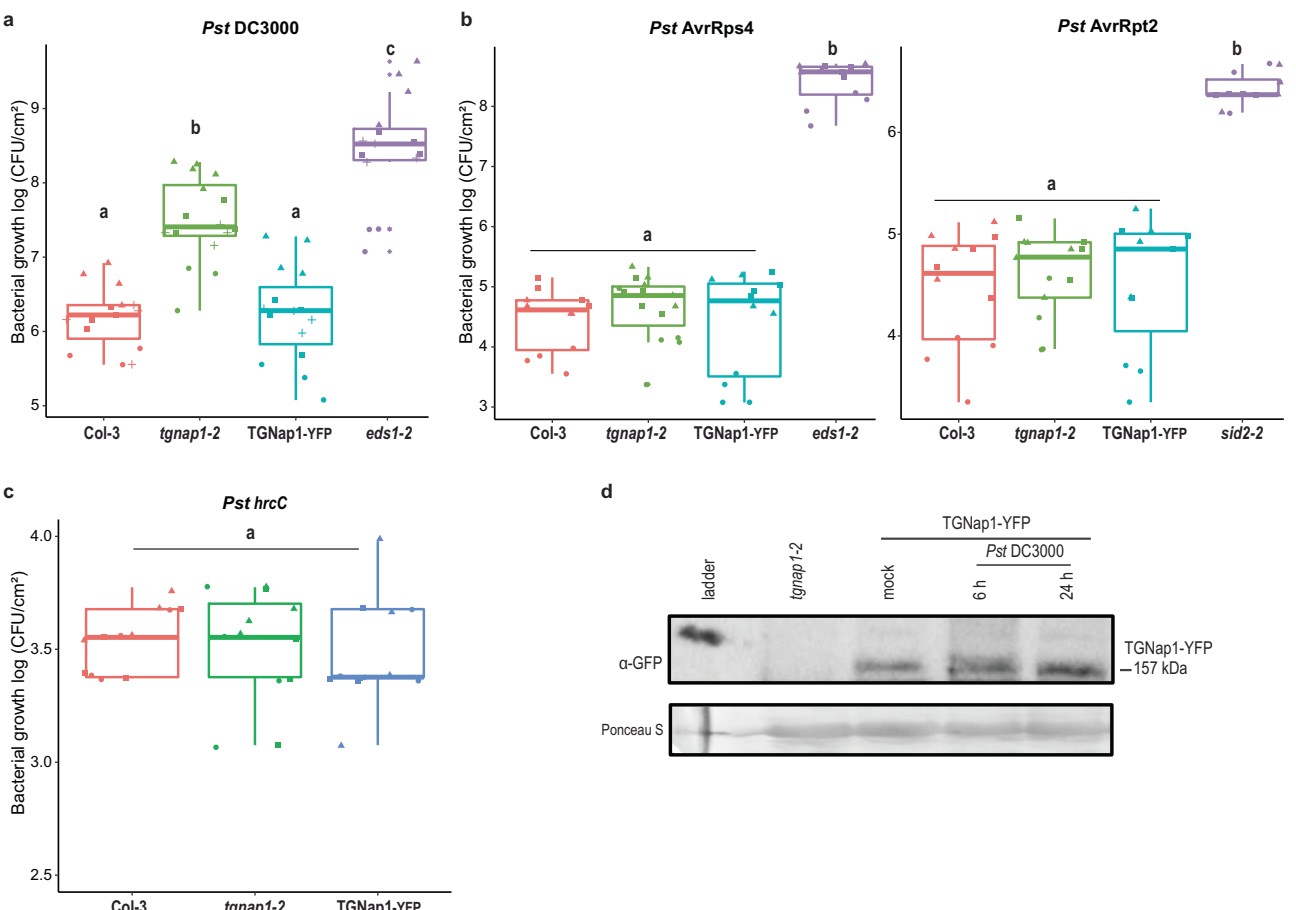

**Fig. 1 | TGNap1 is required for a robust immune response.** Four-week-old *Arabidopsis* plants of the indicated genotypes were infiltrated with **a** *Pst* DC3000, **b** *Pst* AvrRps4 and *Pst* AvrRpt2, **c** *Pst hrcC*. Bacterial titers were determined at 3 dpi. Data from at least three independent experiments are represented in a boxplot with data points (technical replicates). Individual data points are represented with different shapes. Statistical differences between genotypes were analyzed using ANOVA (Tukey's HSD, $P < 0.05$). **d** Accumulation of TGNap1-YFP in four-week-old leaves of TGNap1-YFP transgenic plants treated with mock (10 mM MgCl₂) and *Pst* DC3000 (6 h and 24 h) probed with α-GFP serum. *Tgnap1-2* leaves were used as control to ascertain the specificity of protein bands. Ponceau red was used as loading control. Experiments were repeated twice with similar results.

accumulating SA. We first tested the expression of *Isochorismate synthase 1* (*ICS1/SID2*), a crucial enzyme in SA biosynthesis and marker for SA biosynthesis initiation[27]. In pathogen-unchallenged conditions (mock), we found no differences in *ICS1* expression across Col-3, *tgnap1-2*, and TGNap1-YFP. Furthermore, upon infiltration with *Pst* DC3000, Col-3, *tgnap1-2*, and TGNap1-YFP had a comparable increase in *ICS1* expression (Fig. 2a). Next, we assayed the levels of SA accumulation by infiltrating *Pst* DC3000 into Col-3 and *tgnap1-2*, using *sid2-2* as a control. At 24 h postinfection (hpi) with *Pst* DC3000, we observed an increase in SA accumulation in Col-3 and *tgnap1-2*, but not in *sid2-2* (Fig. 2b). Therefore, because these results support the possibility that the immune defect of *tgnap1-2* is not due to an inability to activate or accumulate SA, we tested if TGNap1 and SA operated in distinct pathways. We generated a *tgnap1-2 sid2-2* double mutant and infected it with *Pst* DC3000. Although *tgnap1-2* accumulated WT-like levels of SA, both *sid2-2* and *tgnap1-2* displayed comparable susceptibility to *Pst* DC3000 (Fig. 2c). Furthermore, the *tgnap1-2 sid2-2* mutant was more susceptible to *Pst* DC3000 than either of the single mutants, indicating that TGNap1 and SA have additive roles in immunity.

Cell-surface receptors are constitutively recycled by endocytosis. Homeostasis of receptors at the PM and recognition of pathogen-associated patterns by such surface receptors activate downstream signaling pathways[37,38]. FLS2, a PM-localized receptor recognizing bacterial flagellin, is endocytosed into endosomes upon sensing flagellin, leading to defense activation[39]. Because

flg22-mediated signaling was not affected in *tgnap1-2* (Supplementary Fig. S1c), we further tested if pre-treating with flg22 would confer protection to plants in the absence of TGNap1. Plants were infiltrated with either mock (DMSO) or flg22 (2 µM), and 24 h later, plants were infiltrated with *Pst* DC3000. We found no differences between untreated (no pre-treatment with mock) and mock-treated plants, whereas *tgnap1-2* showed increased susceptibility to *Pst* DC3000 compared to Col-3 and TGNap1-YFP, as expected (Figs. 1a and 2d). Upon flg22 treatment, Col-3, *tgnap1-2*, and TGNap1-YFP showed significantly decreased bacterial growth compared to mock-treated plants, supporting an induction of resistance (Fig. 2d). As expected, the *fec* mutant had bacterial growth comparable to mock-treated plants (Fig. 2d). Resistance to *Pst* DC3000 was induced in *tgnap1-2* pre-treated with flg22, as evidenced by reduced bacterial growth, phosphorylation of MPKs and increased expression of the marker for actuation of immune responses, *Pathogenesis-related 1* (*PR1*) (Fig. 2d and Supplementary Figs. S1c and S4a). Nonetheless, when pre-treated with flg22, *tgnap1-2* showed higher bacterial growth than Col-3, highlighting that the immunity induced by flg22 in *tgnap1-2* did not alleviate the susceptibility of *tgnap1-2*. To validate these findings, we treated Col-3 and *tgnap1-2* with benzothiadiazole (BTH), a synthetic analog of SA that induces resistance to *Pst* DC3000[40]. To check the efficacy of BTH treatment, we measured *PR1* expression levels, and found them increased upon BTH treatment (Supplementary Fig. S4b). We also found that Col-3 and *tgnap1-2* pre-

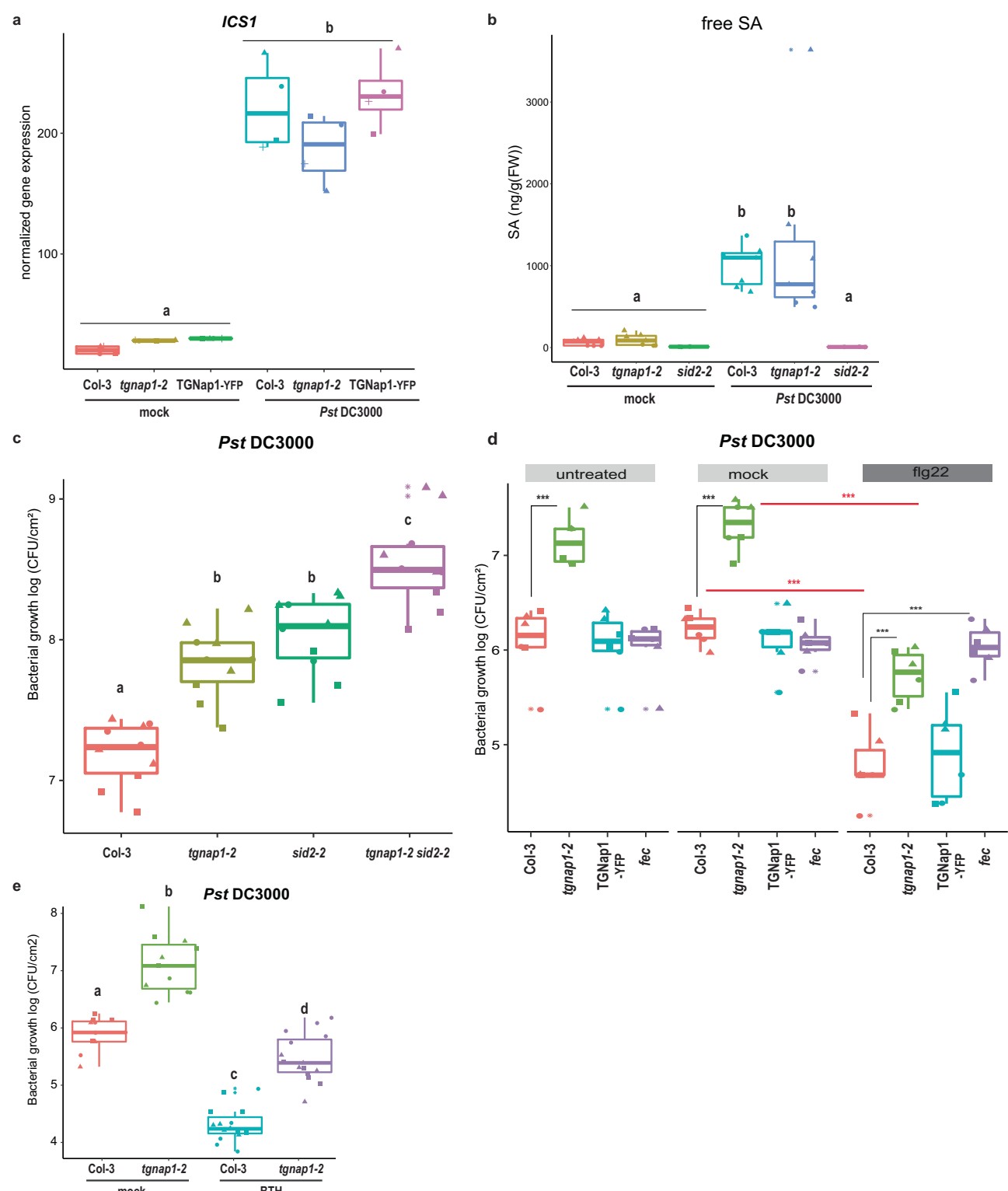

treated with BTH showed reduced bacterial growth compared to mock-treated plants (Fig. 2e). However, even upon immunity induction by BTH, *tgnap1-2* plants allowed increased bacterial growth compared to Col-3 (Fig. 2e), reinforcing that, although immune signaling leading to *PR1* expression remains intact in the *tgnap1-2* mutant, BTH-mediated immunity requires TGNap1. Taken together, these results point to a role of TGNap1 in basal immunity that likely acts at a late stage in immune signaling, downstream of SA.

## TGNap1 is not required for transcriptional reprogramming of canonical defense pathways

To identify signaling pathways affected in *tgnap1-2* plants upon infection, we investigated the transcriptomes of *tgnap1-2* and Col-3 upon *Pst* DC3000 infection. Four-week-old *Arabidopsis* plants were infiltrated with either mock or *Pst* DC3000 and samples harvested at 24 hpi for whole transcriptome sequencing (RNA-seq) analyses. The analyses identified 6206 and 8471 genes differentially expressed

**Fig. 2 | TGNap1 defect in immune signaling is independent of SA immune signaling node and is not alleviated by immune activation. a** Four-week-old *Arabidopsis* plants of the indicated genotypes were infiltrated with either mock or *Pst* DC3000 and samples harvested 24 hpi. *ICS1* gene expression was analyzed with qRT-PCR. Data from three independent experiments are represented in a boxplot. Statistical differences were analyzed using ANOVA (Tukey's HSD, $P < 0.05$). **b** Four-week-old *Arabidopsis* plants of the indicated genotypes were infiltrated with either mock or *Pst* DC3000 and free SA was quantified 24 hpi. Data from two independent experiments are represented in a boxplot. Statistical differences were analyzed using ANOVA (Tukey's HSD, $P < 0.05$). **c** Four-week-old *Arabidopsis* plants of the indicated genotypes were infiltrated with *Pst* DC3000. Bacterial titers were determined at 3 dpi. Data from three independent experiments are represented in a boxplot with data points (technical replicates). Individual data points are represented with different shapes. Statistical differences between genotypes were analyzed using ANOVA (Tukey's HSD, $P < 0.05$). **d** Four-week-old *Arabidopsis* plants of the indicated genotypes were pre-treated by infiltration with either mock or flg22. Twenty-four hours later, plants were infiltrated with *Pst* DC3000. Bacterial titers were determined at 3 dpi. Data from three independent experiments are represented in a boxplot with data points (technical replicates). Individual data points are represented with different shapes. Statistical differences between genotypes (black) or between treatments (red) were analyzed using Student's $t$ test ($P < 0.05$). **e** Four-week-old *Arabidopsis* plants of the indicated genotypes were pre-treated by spraying with either mock or BTH. Twenty-four hours later, plants were infiltrated with *Pst* DC3000. Bacterial titers were determined at 3 dpi. Data from three independent experiments are represented in a boxplot with data points (technical replicates). Individual data points are represented with different shapes. Statistical differences between genotypes were analyzed using ANOVA (Tukey's HSD, $P < 0.05$).

(DEGs) in *Pst* DC3000-treated Col-3 and *tgnap1-2*, respectively, compared to the corresponding mock controls (Fig. 3a). Although we observed a significant number of DEGs upon *Pst* DC3000 infection, the overall expression profile was comparable between Col-3 and *tgnap1-2*, with DEGs being either upregulated or downregulated in a largely overlapping manner in both backgrounds (Fig. 3b). Among the DEGs, 2415 and 3002 DEGs were upregulated and downregulated, respectively, in both Col-3 and *tgnap1-2* compared to mock. Only three genes (*Basic helix-loop-helix23*, *Expansin-like1*, and *Seven in Absentia2*) were upregulated in Col-3 but downregulated in *tgnap1-2*, while four genes (*WRKY DNA-binding protein23*, *Glycosyl hydrolase 9B18*, *Subtilisin-like protease 1.9*, and *AT3G61920*) were downregulated in Col-3 but upregulated in *tgnap1-2* (Fig. 3b and Supplementary Data 1). These gene expression profiles support that the hyper-susceptibility of *tgnap1-2* to *Pst* DC3000 is not due to a defect in transcriptional regulation of specific immune signaling pathways compared to Col-3. From the DEGs, we selected a total of 9257 non-redundant genes (i.e., differentially expressed in at least one genotype) for further analyses. K-means clustering analyses of the 9257 DEGs resulted in the generation of six gene clusters (Fig. 3c): Cluster 1 and 2 with genes more upregulated in *tgnap1-2* than Col-3; Clusters 3 and 5, with genes more downregulated in *tgnap1-2* than Col-3; Cluster 4, with genes downregulated in both *tgnap1-2* and Col-3; Cluster 6, with genes upregulated in both *tgnap1-2* and Col-3 (Fig. 3c). Gene Ontology (GO) term analyses showed strong enrichment of energy-related metabolic pathways in Clusters 3 and 5 (the two top-downregulated DEGs) (Fig. 3d), which indicates enhanced downregulation of genes associated with growth in *tgnap1-2* relative to Col-3. Interestingly, biotic stress-related pathways, including jasmonic acid (JA) and SA signaling, were strongly enriched in Clusters 1 and 2 in which DEGs were more upregulated in *tgnap1-2* relative to Col-3, indicating a hyperactivation of biotic stress-related genes in *tgnap1-2*. Upon comparison of the 9257 non-redundant DEGs with SA- or JA-responsive genes[41,42], we found ~15% overlap with genes grouped in Clusters 1 and 2 (257/1681genes) (Fig. 3e). These results indicate that although GO terms related to SA and JA were enriched in Clusters 1 and 2, the genes in these clusters represent a small portion of the DEGs with enhanced upregulation in *Pst* DC3000-infected *tgnap1-2* compared to Col-3 (Fig. 3e). Therefore, together our data suggest that, during *Pst* DC3000 infection, the absence of TGNap1 leads to a generally increased transcriptional reprogramming with downregulation of growth-related genes and upregulation of biotic stress-related genes, which would be expected for a growth-defense trade-off[43]. Such transcriptional reprogramming does not appear to alter the activation of a specific immune signaling pathway.

## TGNap1 plays a critical role in the secretion of antimicrobial proteins

Because TGNap1 is necessary for exocytic traffic of bulk cargo[15], we hypothesized that TGNap1 could play a role in trafficking of antimicrobial proteins to the apoplast. To test this, we aimed to determine the transcript and protein levels of the antimicrobial protein, PR1, upon *Pst* DC3000 treatment. We found that, upon *Pst* DC3000 treatment, *PR1* transcript levels were induced in Col-3, *tgnap1-2* and TGNap1-YFP infected leaves (Fig. 4a). The total protein levels of PR1 and PR2 (Fig. 4b) were comparable between Col-3, *tgnap1-2* and TGNap1-YFP infected leaves, indicating that the loss of *TGNap1* does not affect the abundance of PR1 and PR2. This is in agreement with canonical SA-mediated defense pathways not being affected in *tgnap1-2* (Figs. 2 and 3). Next, to test whether PR1 and PR2 secretion into the apoplast was affected by the loss of TGNap1, we isolated apoplastic fluids from leaves infiltrated with either mock or *Pst* DC3000. Total protein and apoplastic protein fractions were probed with α-PR1 and α-PR2 specific antibodies. Col-3, *tgnap1-2*, and TGNap1-YFP accumulated comparable levels of total PR1 and PR2 protein, but notably, only *tgnap1-2* displayed reduced PR1 levels in the apoplastic fraction (Fig. 4b and Supplementary Fig. S6). To determine the levels of cytosolic protein contamination during apoplast fluid isolation, the protein samples were probed with antibodies to the cytosolic marker cytosolic fructose-1,6-bisphosphatase (cFBPase). We detected some cFBPase signal at comparable levels across all samples (Fig. 4b), indicating that the differences in the levels of PR1 and PR2 between Col and *tgnap1-2* were not due to cytosolic contamination. Indeed, higher levels of PR1 were found in the intracellular extracts from *tgnap1-2* compared to Col-3 (intracellular, Supplementary Fig. S5d). These results show that TGNap1 is involved in the secretion of PR1 and PR2 to the apoplast upon immune activation.

To test if other proteins were transported to the apoplast in a TGNap1-dependent pathway during immunity, we quantitatively and qualitatively assayed and compared the apoplastic fluid proteome of Col-3 and *tgnap1-2* before and after immune activation. To do this, we isolated apoplastic fluids from *Arabidopsis* leaves infiltrated with either mock or *Pst* DC3000 and subjected it to isobaric peptide labeling (labeling efficiency: mock 94%, and *Pst* DC3000-infiltrated samples 100%), for protein identification by MS/MS. The plant genotypes were paired for comparison based on treatment (i.e., mock or *Pst* DC3000, to account for differences in protein population and abundance). A total of 940 and 922 proteins were detected in mock and *Pst* DC3000 samples, respectively. From the total detected proteins, a cutoff (minimum of two peptides and 1% FDR, Supplementary Data 2) was set to identify differentially abundant proteins. 793 and 786 proteins were detected with this cutoff in the apoplast fluids of mock and *Pst* DC3000 samples, respectively (Supplementary Data 2 and Supplementary Fig. S5). A significant overlap was observed in the apoplastic proteome of mock and *Pst* DC3000-treated Col-3 and *tgnap1-2* plants (Supplementary Fig. S5b). A GO-enrichment analysis of proteins found in both mock and *Pst* DC3000-treated genotypes revealed a significant representation of proteins involved in stress responses (e.g., pathogenesis-related proteins, aspartyl proteases, pectin methylesterases, lipid-transfer proteins (LTP5)) (Supplementary Data 3). The data were further subjected to permutation test ($P < 0.05$ post-hoc:

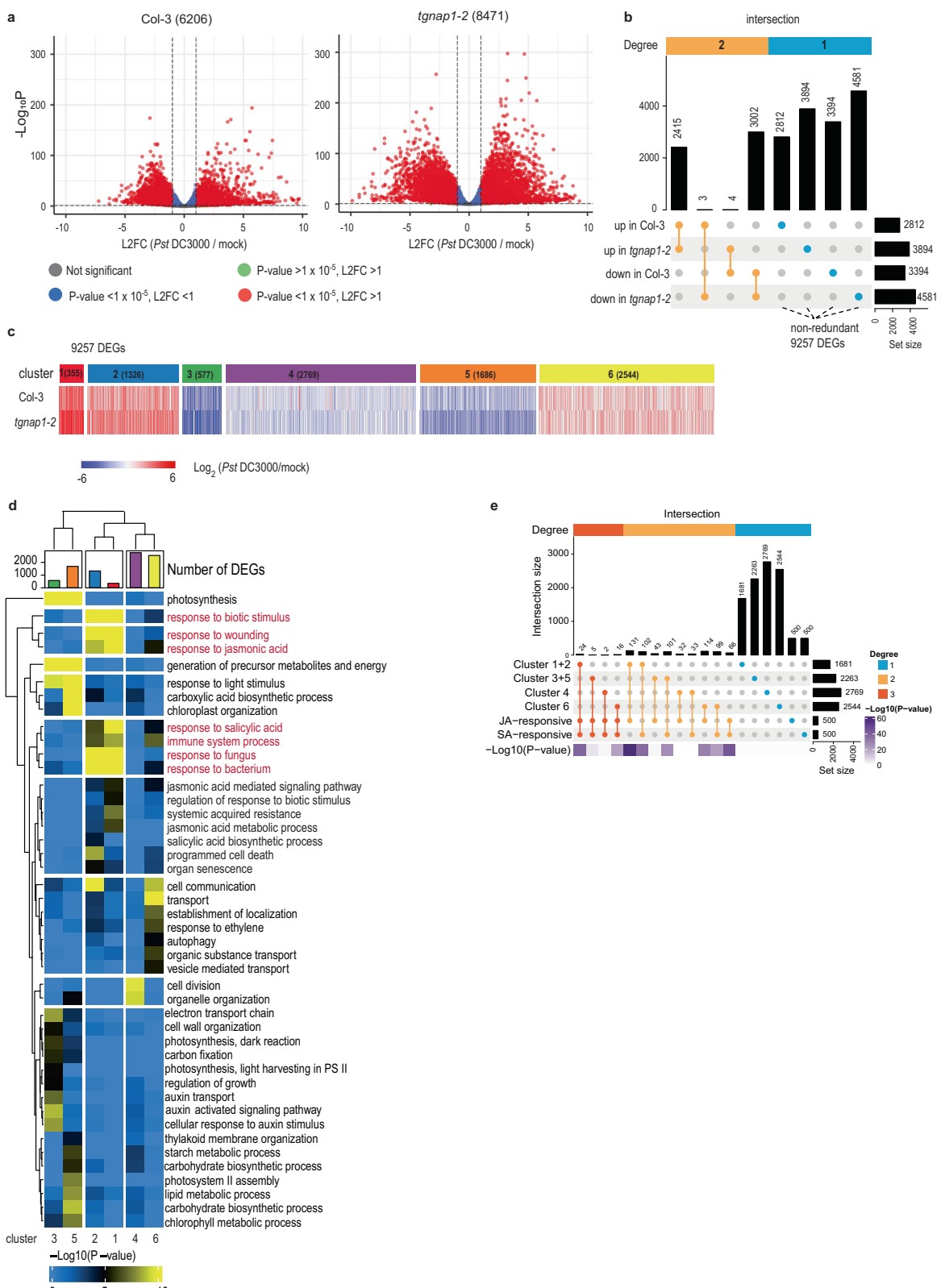

Benjamini–Hochberg) to account for variation between samples and multiple-testing errors. Using a criterion of ±1 fold change, we found that, compared to Col-3, 18 proteins were more abundant in *tgnap1-2* and 7 proteins were less abundant in *tgnap1-2* in *Pst* DC3000-treated plants compared to Col-3 (Supplementary Data 4). The 18 proteins more abundant in *tgnap1-2* relative to Col-3, included proteins involved in biotic stress (e.g., pectin methyl esterase 4 (PME4), lipid-transfer protein 5 (LTP5), glycine-rich protein 3 and glycine-rich protein 3 short isoform) and abiotic stress (e.g., ascorbate oxidase, responsive to desiccation 22, glycosyl hydrolase 17 (GH17)) (Fig. 4c). Interestingly, upon *Pst* DC3000 infection, the proteins less abundant in *tgnap1-2* apoplast compared to Col-3 were exclusively represented by antimicrobial proteins (Fig. 4c and Supplementary Fig. S7) of the pathogenesis-related proteins family (i.e., PR1, PR2, PR5)[44], berberine-

**Fig. 3 | Loss of TGNap1 leads to misregulation of gene expression upon *Pst* DC3000 infection. a** Volcano plots representing global transcriptome changes upon *Pst* DC3000 infection compared to mock at 24 hpi in Col-3 and *tgnap1-2*. The numbers of DEGs are indicated in parentheses. L2FC Log2 fold change. **b** An upset plot representing intersection (overlap) of DEGs between Col-3 and *tgnap1-2*. The dot plot (orange) lists intersections between the indicated datasets. The vertical barplot displays the number of DEGs in each intersection, and the horizontal bar-plot on the right shows the total number of DEGs either upregulated or down-regulated in each genotype. The non-redundant 9257 DEGs were selected for further analyses. **c** A heatmap of expression patterns of 9257 DEGs in Col-3 and

*tgnap1-2*. The DEGs were clustered using a K-means algorithm with Pearson correlation-based distance. Genes were hierarchically clustered according to the expression patterns. **d** A heatmap depicting enrichment of gene ontology terms of the clusters identified in (**c**). The barplot at the top is color-coded as in (**c**) and displays the number of DEGs in each cluster. **e** An upset plot representing the intersection (overlap) of genes in Cluster 1 + 2 (enhanced upregulation), Cluster 3 + 5 (enhanced downregulation), Cluster 4 or Cluster 6 with JA- and SA-responsive genes. The significance of the overlaps between the two datasets or among multi-sets is accessed using Student's *t* test or hypergeometric tests and represented as a heatmap (-Log₁₀P-value).

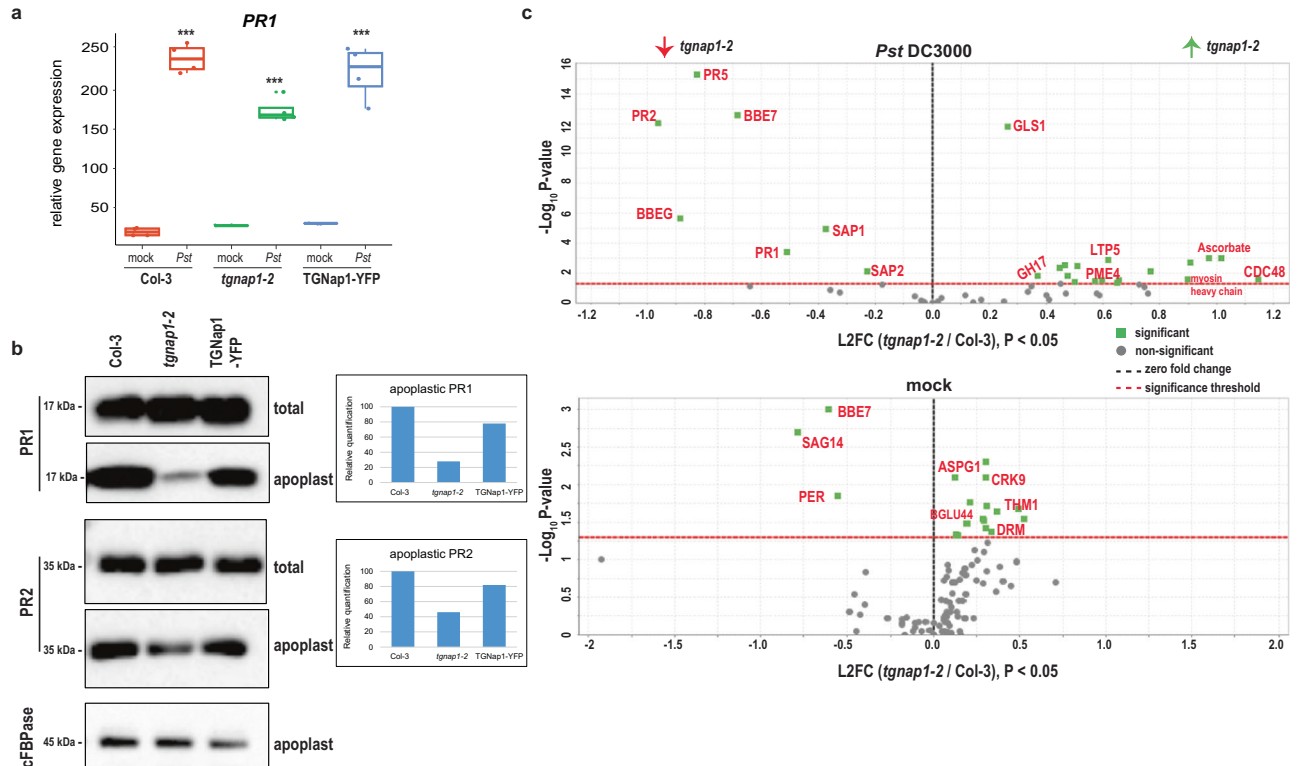

**Fig. 4 | TGNap1 plays a role in exocytic traffic of antimicrobial proteins. a** *PR1* expression in leaves treated with either mock or *Pst* DC3000 (*Pst*) was analyzed with qRT-PCR. Data from three independent experiments are represented in a boxplot. *** indicates significant differences (Student's *t* test) with the respective mock-treated genotypes. **b** Leaf apoplastic fluids were isolated 24 hpi with either mock or *Pst* DC3000. Total and apoplastic protein samples were probed with α-PR1, α-PR2, and α-cFBPase in immunoblot analyses. A quantification of PR1, PR2 protein levels in *Pst* DC3000-treated apoplastic samples, normalized to Col-3 is

represented as a barplot. Experiments were repeated three times with similar results. **c** Leaf apoplastic fluids was isolated 24 hpi with either mock or *Pst* DC3000. Isolated apoplastic fluid were subjected to label-free LC-MS analysis. Proteins detected with at least two peptides with a 1%FDR were selected for analysis. Proteins were filtered with a cutoff of *P* < 0.05 (Benjamini–Hochberg corrected). Selected proteins have been highlighted (for a full list see Supplementary Data 4). L2FC Log2 fold change.

bridge enzymes (BBE4, BBE7), and secreted serine aspartic proteases (SAP1 and SAP2)[45]. Except BBE7, none of the above proteins were differentially abundant in the apoplast of mock-treated Col-3 and *tgnap1-2* plants (Supplementary Fig. S7). These results are consistent with role of TGNap1 in trafficking of apoplastic proteins[15] but also highlight that TGNap1 is required for efficient secretion of antimicrobial proteins upon *Pst* DC3000 infection. Taken together, our results show that during *Pst* DC3000 infection the loss of TGNap1 does not affect immune activation but plays a significant role in the transport of anti-bacterial proteins to the apoplast.

### The interaction of TGNap1 with MTs is required but not sufficient for immunity

MTs are involved in protein secretion to the apoplast, and TGNap1, which binds MTs, is required for the homeostasis of a TGN subpopulation[15]. Therefore, we next aimed to test if disruption of

MTs would influence TGNap1-mediated transport of PR1 to the apo-plast. To do so, Col-3, *tgnap1-2*, and *eds1-2* were co-infiltrated with *Pst* DC3000 and oryzalin, which depolymerizes MTs. Control plants were co-infiltrated with *Pst* DC3000 + DMSO (oryzalin solvent). Apoplastic fluids were isolated 24 hpi and subjected to immunoblot analyses with α-PR1 and cFBPase (Fig. 5a). Col-3 plants treated with *Pst* DC3000 showed apoplastic accumulation of PR1, while the *eds1-2* mutant showed undetectable PR1 accumulation, as expected (Fig. 5a). Except for *eds1-2*, a significant cytosolic protein signal was not observed in the samples (Fig. 5a), indicating that the difference in apoplastic PR1 may not be attributed to cytosol contamination. When Col-3 plants were co-infiltrated with *Pst* DC3000 + oryzalin, a significant reduction in PR1 accumulation in the apoplast occurred compared to Col-3 infiltrated with *Pst* DC3000 only. These results indicate that PR1 secretion in WT largely depends on MT integrity and TGNap1 availability. We next assayed the role of TGNap1 for the

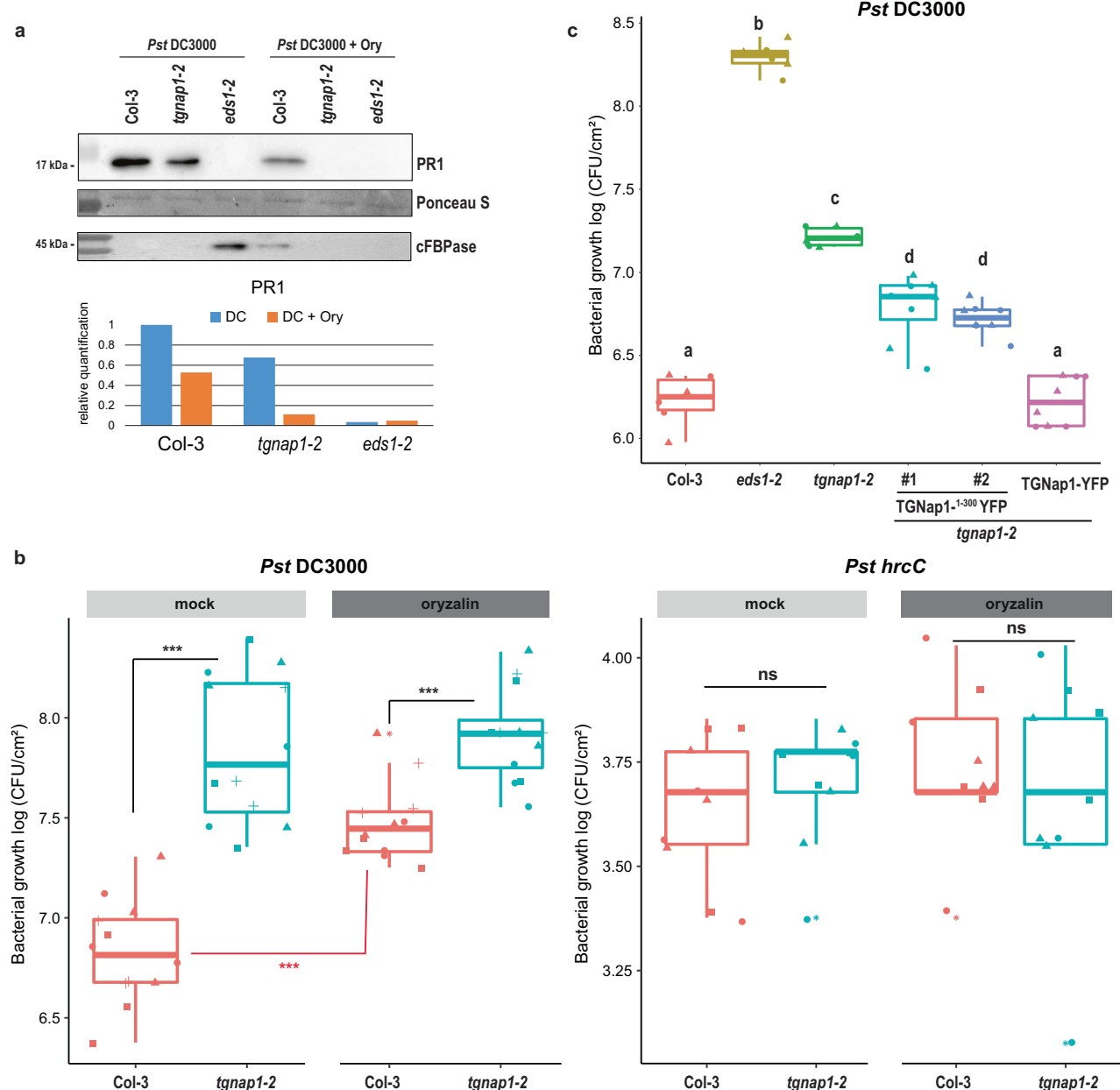

**Fig. 5 | Loss of TGNap1 mimics oryzalin-mediated MT disruption during *Pst* DC3000 infection. a** Leaf apoplastic fluid was isolated 24 hpi with either *Pst* DC3000 + mock or *Pst* DC3000 + oryzalin (Ory). Samples were probed with α-PR1 in immunoblot analyses. A quantification of PR1 protein levels in *Pst* DC3000-treated apoplastic samples, normalized to Col-3, is represented in a barplot. Experiments were repeated three times with similar results. **b** Four-week-old *Arabidopsis* plants of Col-3 and *tgnap1-2* were co-infiltrated with either *Pst* DC3000 + 10 mM MgCl2 (mock) or *Pst* DC3000 + oryzalin (oryzalin) or *Pst hrcC* + 10 mM MgCl2 (mock) or *Pst hrcC* + oryzalin (oryzalin). Bacterial titers were determined at 3

dpi. Data from three independent experiments are represented in a boxplot with data points (technical replicates). Individual data points are represented with different shapes. Statistical differences between genotypes (black) or between treatments (red) were analyzed using Student's *t* test (*P* < 0.05). **c** Four-week-old *Arabidopsis* plants of the indicated genotypes infiltrated with *Pst* DC3000. Bacterial titers were determined at 3 dpi. Data from two independent experiments are represented in a boxplot with data points (technical replicates). Individual data points are represented with different shapes. Statistical differences between genotypes were analyzed using ANOVA (Tukey's HSD, *P* < 0.05).

secretion of PR1 in conditions of MT depletion. In agreement with our apoplast fluid proteomic analyses and western blot analyses (Fig. 4b, c), *tgnap1-2* showed a significantly reduced PR1 accumulation in the apoplast compared to Col-3. Notably, when *tgnap1-2* plants were co-infiltrated with *Pst* DC3000 + oryzalin, barely any PR1 was detectable in the apoplast compared to *tgnap1-2* infiltrated with *Pst* DC3000 only (Fig. 5a). Because MT disruption enhances pathogen growth by providing a virulence advantage to bacteria[17], we tested if the reduced PR1 secretion in *tgnap1-2* plants would further increase pathogen growth. To test this, we measured bacterial

growth in Col-3 and *tgnap1-2* plants with co-infiltrated *Pst* DC3000 + oryzalin. Co-infiltration with oryzalin led to increased bacterial growth in Col-3, compared to plants infiltrated only with *Pst* DC3000 (Fig. 5b), supporting an increase in susceptibility due to disrupted MTs. Remarkably, disruption of MTs, which massively reduced apoplastic PR1 levels (Fig. 5a), did not result in increased susceptibility in *tgnap1-2* (Fig. 5b). The oryzalin-induced increase in susceptibility was limited to effector-producing *Pst* bacteria, as *Pst hrcC* did not show increased growth in oryzalin-treated plants compared to mock in both Col-3 and *tgnap1-2* (Fig. 5b), indicating that bacterial effectors

may disrupt MTs to further dampen immune responses. To control for the effect of MT disruption on immunity, we co-infiltrated *Pst* DC3000 with the MT-polymerizing chemical taxol[46]. In contrast to oryzalin, *Pst* DC3000 co-infection with taxol had no effect on bacterial growth with the mock and taxol-treated plants being indistinguishable in bacterial titers (Supplementary Fig. S8). Notably, oryzalin-mediated MT disruption or taxol-mediated polymerizing did not provide a virulence advantage in *tgnap1-2*, suggesting that any advantage provided by MT disruption would overlap with the loss of *TGNap1*. However, the decreased PR1 levels in *tgnap1-2* plants treated with oryzalin (Fig. 5a) suggest that TGNap1 also contributes to PR1 traffic in MT-independent pathways.

To further understand the TGNap1 immune function/s at MTs, we next aimed to test if MT binding of TGNap1 was sufficient to recover susceptibility in *tgnap1-2*. The first 182 aa of TGNap1 are sufficient to bind MTs[15]. The MT-binding region of TGNap1 is followed by two predicted coiled-coil regions (region 1: aa 215–249; region 2: aa 257–291), which might affect the structural integrity or folding of TGNap1. To avoid improper protein folding, we generated a truncated TGNap1 protein comprising of its MT-binding domain and a predicted coiled-coil domain and fused to YFP (TGNap1$^{1-300}$-YFP), for transformation into *tgnap1-2*. Two independent TGNap1$^{1-300}$-YFP transgenic lines were tested for resistance against *Pst* DC3000. Col-3 and full-length TGNap1-YFP lines showed comparable bacterial growth, while *tgnap1-2* and *eds1-2* were hypersusceptible (Fig. 5c). The truncated TGNap1$^{1-300}$-YFP lines showed intermediate susceptibility with bacterial levels between Col-3 and *tgnap1-2*, indicating only a partial recovery of TGNap1 immune functions (Fig. 5c). Together these results reinforce that MT binding is required but not sufficient for TGNap1's functions in immunity.

## Identification of TGNap1 highlights a functional diversity of TGN proteins in immunity

The plant TGN is highly heterogeneous and made of partially overlapping populations. Because TGNap1 is necessary for the MT-dependent biogenesis of a TGN subpopulation[15], we aimed to test if the verified role of TGNap1 in immunity could overlap with other TGN-associated proteins involved in immunity. MIN7 is a TGN-localized ARF-GEF protein playing an important role in PTI, ETI, and pathogen-activated SA responses[4,23]. MIN7 is postulated to have a role in vesicle trafficking, based on reduced callose deposition upon pathogen infection[47]. Therefore, we co-expressed TGNap1-YFP, MIN7-mCherry, and the TGN marker SYP61-CFP[15]. We verified that the signals of these markers largely overlapped (Fig. 6a, b). Using Pearson's correlation coefficient analyses[48], we found that the average signal overlap between TGNap1/SYP61 was 77.5% (Fig. 6b), in accordance with a partial co-distribution of these proteins at the TGN[15]. When we analyzed the distribution of MIN7 with respect to SYP61 and TGNap1. We found that the Pearson's correlation for the signal overlap for TGNap1/MIN7 and SYP61/MIN7 was 63.9% and 64.9% (Fig. 6b), respectively, highlighting a non-complete overlap of the distribution of MIN7 with SYP61 and TGNap1 at a subcellular level. Indeed, we observed distinct punctate structures with MIN7, which did not contain either TGNap1 or SYP61 signal (Fig. 6a, c, arrowheads). Conversely, we also verified TGN populations where the fluorescent protein fusions to TGNap1, MIN7, and SYP61 co-localized (Fig. 6c, arrow; Fig. 6d). These results indicate that, while TGNap1 and MIN7 co-localize at the same TGNs, they also localize to distinct TGN subsets, which could underscore some degree of both functional redundancy and functional diversity. Therefore, to further investigate if TGNap1 and MIN7 could have overlapping and/or distinct roles in immunity, we generated a *tgnap1-2 min7* double mutant that we used to quantify bacterial growth for comparison to WT, *min7*[47], and *tgnap1-2*. Upon infection with *Pst* DC3000, we found that *min7* was slightly more susceptible than Col-3 while *tgnap1-2* showed higher bacterial growth than both Col-3 and *min7* (Fig. 6e). The

*tgnap1-2 min7* double mutant was more susceptible than either of the single mutants indicating that TGNap1 and MIN7 are involved in separate immune pathways. MIN7 is degraded by the *Pst* DC3000 effector HopM1, and the *min7* mutant is more susceptible to *Pst* ΔEM (a *Pst* DC3000 mutant lacking HopM1) compared to WT[47]. To further understand the immune functions of TGNap1 and MIN7, we infected *tgnap1-2 min7* with *Pst* ΔEM. Upon infection with *Pst* ΔEM we observed that *tgnap1-2* was more susceptible than Col-3 (Supplementary Fig. S2b and Fig. 6f), but less susceptible than *min7*, which had 5-fold increased bacterial growth than *tgnap1-2* (Fig. 6f). The *tgnap1-2 min7* double mutant was more susceptible to *Pst* ΔEM than either of the single mutants, further indicating that, although the TGNs marked by TGNap1 and MIN7 partially coincide, these proteins have partially overlapping roles in immunity. In summary, our work identifies TGNap1 as a MT-binding and TGN-associated protein with a critical role in plant immunity.

## Discussion

Robust plant immune signaling depends on early (transcriptional reprogramming) and late responses, the latter encompassing steps necessary to secrete antimicrobial proteins in the apoplast. Although a wealth of information on proteins involved in early responses, which include pathogen recognition and mechanisms underlying activation of immune signaling cascades, is available[1,3,6,49], the identity of proteins involved in trafficking of antimicrobial proteins to the cell surface is still largely unknown. In this work, we identify TGNap1 as a key player in the efficient transport of antimicrobial proteins upon *Pst* DC3000 infection. TGNap1 is a TGN-associated protein important for the MT-dependent biogenesis and movement of a TGN-subpopulation[15]. In this work, we demonstrate that TGNap1 is necessary for the transport sufficiency of antimicrobial proteins to the apoplast during immunity rather than for the homeostasis or activation of immune signaling pathways, including the canonical SA-defense pathway. We further show that TGNap1 operates in a defense pathway that only partially overlaps with the immune pathway dependent on TGN-localized MIN7. Based on our results, we propose that TGNap1 is a key element of a MT-dependent pathway necessary for the homeostasis of antimicrobial cargo traffic in late immune responses.

The coevolution of plants and phytopathogens has led to defense proteins and pathways that partially overlap at a functional level[50–52]. In our work, we established that TGNap1's role is specific to basal defense against virulent *Pst* DC3000 and *Hpa* NOCO2, but not in PTI (*Pst hrcC*) and ETI (*Pst* AvrRps4 and *Pst* AvrRpt2). While MPK signaling cascades are not compromised in *tgnap1-2*, TGNap1 is not completely dispensable for PTI as flg22-activated immunity is still compromised in *tgnap1-2*, underscoring a requirement of TGNap1 in different aspects of basal immunity. Activation of ETI, compared to virulent *Pst* DC3000, results in a faster transcriptional reprogramming and is often accompanied by cell death[7,53], which most likely underlies the WT-like resistance of *tgnap1-2* against ETI-activating *Pst* strains. Although we cannot exclude that TGNap1 may be the target of a yet-untested bacterial or Oomycete effector, we found that the *tgnap1-2* mutant supported significant increase in bacterial growth against different *Pst* effector deletion mutants of varying virulence. In addition, our evidence that the transcription and protein levels of TGNap1 were not affected upon infection by *Pst* DC3000 indicates that TGNap1 is unlikely a transcriptional regulator in immunity or a target of a bacterial effector. This is further supported by the evidence for increased bacterial growth in *tgnap1-2* even after immune induction by flg22 or BTH treatments.

Mutants with increased susceptibility to pathogens are usually defective in transcriptional activation of one or more hormonal pathways[53,54]. A prime example is the increased susceptibility to *Pst* DC3000 in mutants defective in SA upregulation or accumulation[27,55]. Significantly, the removal of multiple hormonal pathways allows for

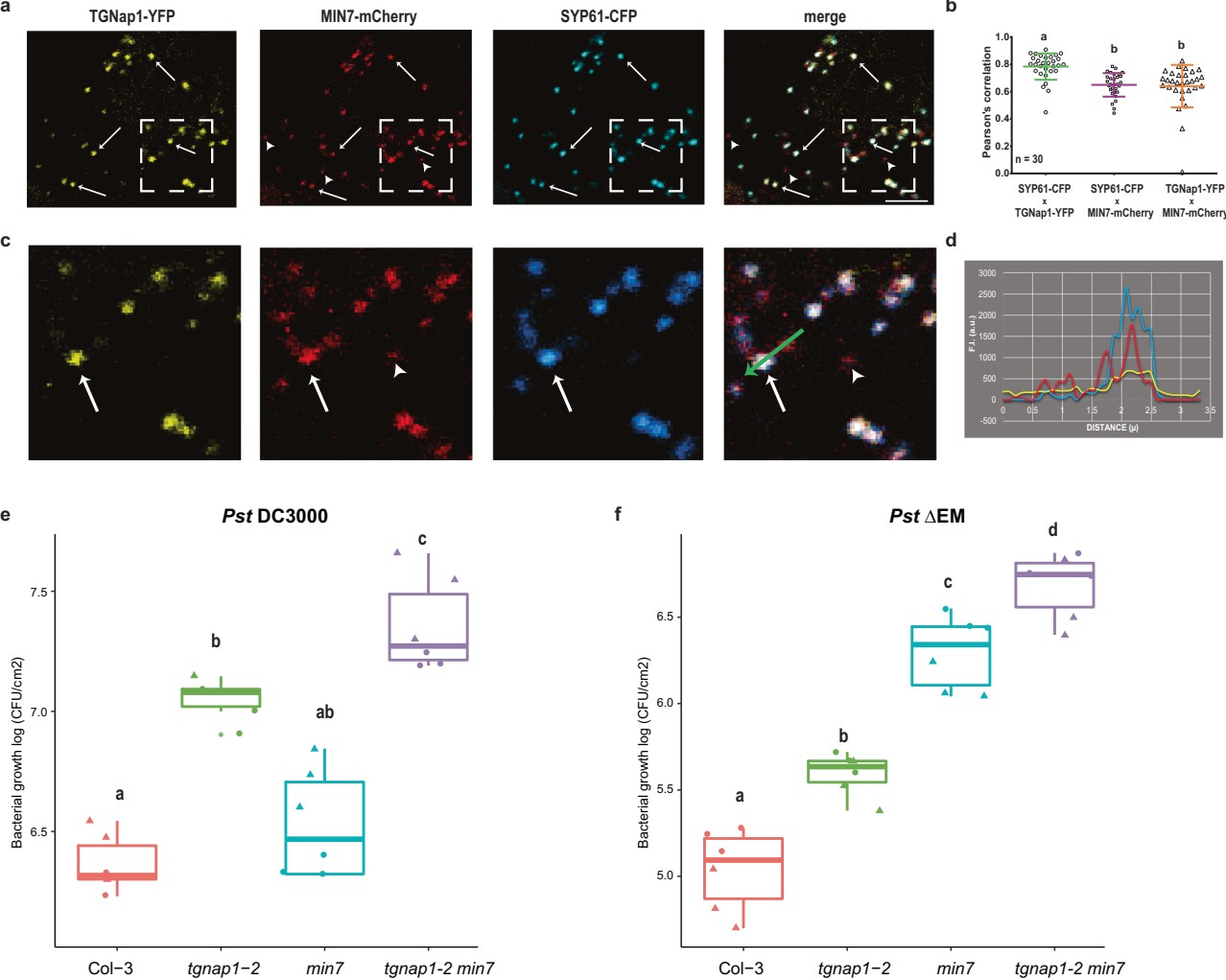

**Fig. 6 | TGNap1 and MIN7 have additive immune signaling roles. a** Confocal images of transiently expressed TGNap1-YFP, MIN7-mCherry, and SYP61-CFP in *N. tabacum* showing TGN localization of TGNap1 and MIN7 with the TGN marker SYP61. Arrows**:** TGNs positive for TGNap1-YFP, MIN7-mCherry, and SYP61-CFP together; arrowheads**:** TGNs positive for MIN7-mCherry but not TGNap1-YFP and SYP61-CFP. Scale bar: 5 μm. Experiments were repeated twice (*n* = 30 images). **b** The degree of co-localization between protein pairs as measured by Pearson's coefficient is represented as a graph. Statistical differences between co-localized proteins were analyzed using ANOVA (*n* = 30 images, Tukey's HSD, *P* < 0.05). **c** Zoom: ×5 magnification of the region marked (white dashed box) in (**a**). **d** Intensity profile (F.I.) measurements for TGNap1-YFP (yellow), MIN7-mCherry (red), and SYP61-CFP (blue) along the arrow (green) on individual TGNs indicating MIN7-specific signal

and overlap between all three signals. **e** Four-week-old *Arabidopsis* plants of the indicated genotypes were infiltrated with *Pst* DC3000. Bacterial titers were determined at 3 dpi. Data from two independent experiments are represented in a boxplot with data points (technical replicates). Individual data points are represented with different shapes. Statistical differences between genotypes were analyzed using ANOVA (Tukey's HSD, *P* < 0.05). **f** Four-week-old *Arabidopsis* plants of the indicated genotypes were infiltrated with *Pst* ΔEM (*Pst* effectors AvrE and HopM1 deletion mutant). Bacterial titers were determined at 3 dpi. Data from two independent experiments are represented in a boxplot with data points (technical replicates). Individual data points are represented with different shapes. Statistical differences between genotypes were analyzed using ANOVA (Tukey's HSD, *P* < 0.05).

residual immunity[54], supporting the existence and activity of non-hormonal mechanisms underlying defense in plants. Using RNA-seq, we established that *tgnap1-2* does not exhibit defect in transcriptional markers of known specific immune signaling pathways. Furthermore, although SA plays a prominent role in defense against *Pst* DC3000, *tgnap1-2* displayed WT-like *ICS1* expression and SA accumulation but was as hypersusceptible to *Pst* DC3000 infection as *sid2-2*. Therefore, we propose that mechanisms downstream of SA accumulation may be compromised by the loss of *tgnap1*. The evidence that canonical defense pathways were not affected by the loss of TGNap1 suggests a late requirement of TGNap1 in immunity, most likely to transport antimicrobial proteins. This is also supported by the fact that a *tgnap1-2 sid2-2* double mutant was more susceptible than either single mutant.

A stringent requirement for TGNap1 in the homeostasis of post-Golgi traffic in physiological conditions of growth has been previously

shown using an artificial bulk flow reporter[15]. The significant protein overlap between pathogen-challenged and unchallenged conditions supports a role of TGNap1 in bulk secretion. However, the dispensable role of TGNap1 in *Pst hrcC* infection and ETI demonstrated in our work supports that TGNap1 may be required also for the secretion of selected cargo in specific conditions of pathogen infection. Indeed, using rigorous statistical filtering, of the proteins differentially represented in the *tgnap1-2* apoplast compared to WT upon *Pst* DC3000 infection, proteins such as PR1, PR2, PR5, and SAP1 and SAP2, were found to be less abundant in the apoplastic fraction of *tgnap1-2* compared to WT. Our results demonstrate that TGNap1 is preferentially required for the exocytosis of anti-bacterial proteins in pathogen-challenged conditions. The reduced abundance of these antimicrobial proteins likely underlies the increased susceptibility of *tgnap1-2* to *Pst* DC3000. Notably, the loss of *TGNap1* did not completely abolish the

exocytic trafficking of these proteins highlighting that other traffic components might be redundantly involved in transporting anti-microbial proteins to the apoplast. In accordance with a role of TGNap1 in endocytic trafficking, a significant number of proteins (e.g., PME4, LTP5, GH17) were also found to be accumulated at higher levels in the apoplast of *tgnap1-2* compared to WT in *Pst* DC3000-treated samples, pointing to a possible role in maintaining protein populations upon infection. However, the perception and likely endocytosis of the flg22-recognizing protein FLS2 was not affected in *tgnap1-2* as evidenced by the flg22-protection assay and MPK phosphorylation assay. While a role of disrupted endocytic trafficking in the susceptibility of *tgnap1-2* cannot be excluded, our results underscore a significant role of TGNap1 in the secretion of antimicrobial proteins in response to pathogen infection and thus identify a component of the exocytic traffic during defense against pathogens.

MTs have an essential role in structural support, developmental processes, vesicle transport, and defense against pathogens[4,56,57]. The cortical MT network is required for targeted exocytosis[15,58]. This makes the plant MT network a target for bacterial effectors for dampening immune response. The *Pst* effector HopZ1 acetylates MTs leading to their disruption, while another *Pst* effector HopE1 dissociates the MICROTUBULE-ASSOCIATED PROTEIN 65-1 (MAP65-1) from MTs[17,59]. It has been recently established that TGNap1 acts as a linker between MTs and TGN, thereby facilitating the movement of TGNs and asso-ciated cargo on MTs[15]. We show here that the bacterial growth in WT plants co-treated with the MT-depolymerizing oryzalin is comparable to *tgnap1-2*. Co-infiltration of *Pst* DC3000 with oryzalin in *tgnap1-2* did not lead to additional susceptibility to *Pst* DC3000, underscoring that MT-mediated pathways are already compromised in *tgnap1-2* and further depolymerizing MTs does not provide additional advantages for bacteria to growth. In addition, taxol-polymerized MTs did not recover the susceptibility of *tgnap1-2* further supporting a MT-homeostasis-dependent role of TGNap1 in immunity. Significantly, apoplastic PR1 accumulation was not detected in the *tgnap1-2* mutant when co-treated with *Pst* DC3000 and oryzalin, which is indicative that TGNap1 contributes to the trafficking of PR1 also independently from MTs. It remains unknown if the role of TGNap1 in immunity is due to its MT-binding ability and/or disruption of TGN-mediated traffic or a combination of both. Complementation of *tgnap1-2* with first 300 amino acids, containing the MT-binding domain (TGNap1[1-300]-YFP) domain partially recovers susceptibility, suggesting a requirement of both the MT binding and TGN/cytosolic TGNap1 for proper function-ing of TGNap1 in immunity.

The TGN is a hub for protein secretion, and in plant cells the TGN is highly heterogeneous with subpopulations of TGNs with different sizes and composition. The loss of TGN integrity either in the number or size hampers secretion[60]. The TGN mutant *syp42 syp43* leads to altered TGN morphology, reduced plant growth, and decreased resistance to *Erysiphe pisi*[60]. On the contrary, loss of ECHIDNA (ECH), another TGN protein, leads to hyperaccumulation of SA, increased expression of defense-marker genes, and enhanced post-invasive resistance to powdery mildew[22]. It has been shown that the loss of *TGNap1* results in clustering of TGNs, and a reduced budding of TGNs from pre-existing TGNs structures[15]. However, unlike the dwarf phe-notype of *syp42 syp43,* and *ech*, at a phenotypic level in physiological conditions of growth, *tgnap1-2* is indistinguishable from WT although it is susceptible to *Pst* DC3000. Furthermore, MIN7, a TGN-localized ARF-GEF protein, is targeted by the bacterial effector HopM1 to dam-pen PTI, and also plays a key role in ETI[23,47]. Using a *tgnap1-2 min7* mutant, which displays increased susceptibility to *Pst* DC3000 and *Pst* ΔEM, we established that TGNap1 and MIN7 have partially distinct roles in immunity. Significantly, TGNap1 and MIN7 partially co-localize to the TGN subpopulations; However, a *tgnap1-2 min7* double mutant is more susceptible than the single mutants highlighting the functional com-plexity of TGN-localized immune proteins. Therefore, the relevance and contribution of the various TGN-associated proteins to TGN integrity, post-Golgi membrane traffic and pathogen responses high-lights a marked functional diversification of the evolution of plant-pathogen interactions. The plant TGN components and TGN sub-populations discovered to date most likely contribute differently to various biotic stress responses. A diversification of plant TGNs could potentially amplify the routes for delivering antimicrobial proteins to the apoplast, which would be advantageous to the host in the event of being hijacked by different pathogens.

## Methods

### Plant materials, growth conditions, and pathogen strains
The mutants *tgnap1-2*, *eds1-2*, *sid2-2,* and *min7* were previously described[15,23,61]. *Pseudomonas syringae pv*. tomato (*Pst*) strain DC3000, *Pst hrcC-* and *Pst* DC3000 effector mutants were used[5]. Plants were grown on soil in controlled environment chambers under a 10 h light regime (120–150 μE/m$^2$s) at 22 °C and 60% relative humidity.

### Pathogen infection assays
For bacterial growth assays, *Pst* (OD$_{600}$ = 0.0005) in 10 mM MgCl$_2$ were hand-infiltrated into leaves of four-week-old plants and bacterial titers measured, as described earlier[61]. Each biological replicate comprised of three leaf discs from different plants, and data shown in each experiment is compiled from 3 to 4 biological replicates. Statistical analyses were performed using one-way ANOVA with post-hoc multi-ple-testing correction using Tukey's HSD ($P < 0.05$).

For bacterial growth assays, *Pst* (OD$_{600}$ = 0.2 + 0.04% silwet L-77) in 10 mM MgCl2 was sprayed onto leaves of four-week-old plants and bacterial titers measured. Each biological replicate comprised of three leaf discs from different plants, and data shown in each experiment is compiled from 3 to 4 biological replicates. Statistical analyses were performed using one-way ANOVA with post-hoc multiple-testing cor-rection using Tukey's HSD ($P < 0.05$).

*Hpa* NOCO2 were sprayed onto 2–3-week-old plants at $5 \times 10^4$ spores/ml in distilled water. To quantify *Hpa* sporulation on leaves, two pots of each genotype were infected, and leaves were sampled from each pot. Plants were harvested at 5 dpi, their fresh weight determined, and conidiospores suspended in 2 ml of distilled water and 10 μl counted with a microscope using a Neubauer counting chamber.

### Chemical and elicitor treatments
For flg22 assays, four-week-old plants were infiltrated with either mock (DMSO) or 2 μM flg22 (purified peptide), after 24 h *Pst* DC3000 was (OD$_{600}$ = 0.0005) hand-infiltrated and bacterial titers measured. Each biological replicate comprised of three leaf discs from different plants, and data shown in each experiment is compiled from 2 to 3 biological replicates.

For BTH assays, four-week-old plants were sprayed with either mock (10 mM MgCl$_2$) or 300 μM BTH (Syngenta), after 24 h *Pst* DC3000 (OD$_{600}$ = 0.0005) was hand-infiltrated and bacterial titers measured. Each biological replicate comprised of three leaf discs from different plants, and data shown in each experiment is compiled from 2 to 3 biological replicates.

For oryzalin and taxol assays, four-week-old plants were co-infiltrated with either mock or chemicals at the indicated concentra-tion with *Pst* DC3000 (OD$_{600}$ = 0.0005) hand-infiltrated and bacterial titers measured at 3 dpi. Each biological replicate comprised of three leaf discs from different plants, and data shown in each experiment is compiled from a minimum of three biological replicates.

### MPK phosphorylation assay
For MPK phosphorylation assays, flg22-treated leaves were harvested at 15 and 30 min after treatment. Samples were snap frozen in liquid nitrogen and homogenized in protein extraction buffer (100 mM

HEPES- pH 7.5, 10% glycerol, 5 mM EDTA, 2 mM dithiothreitol, 1 mM phenylmethylsulfonyl fluoride, and proteinase inhibitor cocktail and phosphatase inhibitor cocktail. Resuspended samples were centrifuged at 130,000 rpm for 10 mins at 4 °C. Protein concentration was determined using Bradford assay. Ten micrograms of protein were used for probing on a western blot using anti-phospho-p44/42 MAPK (1:5000, Cell Signaling Technology, Danvers, MA, USA) as primary antibody, and HRP-conjugated goat anti-rabbit IgG (1:10000, A 6154; Sigma).

## Protein extraction, immunoblotting and quantification

Total leaf extracts were processed in extraction buffer (50 mM Tris pH 7.5, 150 mM NaCl, 10% (v/v) glycerol, 2 mM EDTA, 5 mM DTT, a protease inhibitor (Roche, 1 tablet per 50 ml, 0.1% Triton). Lysates were centrifuged for 15 min, 12,000 rpm at 4 °C. Supernatant after centrifugation were boiled at 95 °C in 2× Laemmli buffer for 10 min. Proteins were separated by SDS-PAGE and analyzed by immunoblotting. Protein bands were quantified using the Bio-Rad Image lab software. Proteins band intensities were normalized to the wild-type protein band within each blot.

## RNA isolation, library preparation, and analysis

Plants were infiltrated with Pst DC3000 ($OD_{600}$ = 0.005). To randomize samples and reduce variation, total RNA was isolated from four individual plants per genotype (three infected leaves per plant) and pooled as one biological replicate. Total RNA was purified with a RNeasy Plant Mini Kit (Qiagen) according to the manufacturer's instructions. RNA-seq libraries were generated using the Illumina TruSeq Stranded mRNA Library (Illumina, San Diego, CA, USA) and sequenced in single-end mode on the Illumina HiSeq 4000 platform (50-nt) at the Research Technology Support Facility Genomics Core at Michigan State University. The quality of raw reads was evaluated using FastQC (version 0.11.5). Reads were cleaned for quality and adapters with Cutadapt (version 1.16)[62] using a minimum base quality of 20 retaining reads with a minimum length of 30 nucleotides after trimming. Quality-filtered reads were aligned to the Col-0 reference genome (TAIR10) using Bowtie (version 2.2.3)[63] and TopHat (version 2.1.1)[64] with a 10-bp minimum intron length and 15,000-bp maximum intron length. Fragments per kilobase exon model per million mapped reads (FPKM) were calculated using TAIR10 gene model annotation with Cufflinks (version 2.2.1). Per-gene read counts were measured using HTSeq (version 0.6.1p1)[65] in the union mode with a minimum mapping quality of 20 with stranded=reverse counting. Differential gene expression analysis was performed in each sample relative to the mock control using DESeq2 (version 1.36.1)[66] within R (version 4.1.3). Genes of which the total count across treatments and replicates in each genotype is <100 were not included in the analysis. All genes analyzed were visualized for each genotype in volcano plots using R package EnhancedVolcano. DEGs were obtained based on adjusted $P$ value < 0.05 and absolute Log2FC > 1. GO-enrichment analysis was performed using agriGO (version 2.0) (http://systemsbiology.cau.edu.cn/agriGOv2/)[67] with a false-discovery rate adjusted $P$ < 0.05 (hypergeometric test with Bonferroni correction) as a cutoff. Biological process GO categories were analyzed and visualized in the heatmap using R package ComplexHeatmap (version 2.14.0)[68]. K-means clustering analysis of the 9257 DEGs was performed with Log2FC outputs generated from DESeq2 using R package factoextra (version 1.0.7).

## qRT-PCR analysis

Total RNA was extracted using a Plant RNA extraction kit (Macherey-Nagel). Five hundred nanograms of total RNA were used for cDNA synthesis (iScript), and qRT-PCR analyses were performed using SYBR green master mix. The housekeeping genes, *GapDH*, were used as reference. Primer efficiencies were above 90% for all oligos, and data

was analyzed by dCt to calculate relative expression. Primers used are listed in Supplementary Table S1.

## SA measurements

Four-week-old plants were infiltrated with *Pst* DC3000 ($OD_{600}$ = 0.005), and samples were harvested 24 hpi. Each biological replicate comprised of six leaves pooled from different plants, and data shown in each experiment is compiled from three biological replicates. A minimum of 100 mg (fresh weight) of flash-frozen leaf tissue was ground using a Retsch mill. SA was extracted overnight using 500 μL of cold extraction buffer (methanol:water (80:20 v/v), 0.1% formic acid, 0.1 g L − 1 butylated hydroxytoluene, 100 nM ABA-d6). Filtered extracts were quantified using an Acquity Ultra Performance Liquid Chromatography (UPLC) system (Waters Corporation, Milford, MA).

## Molecular cloning

A truncated entry vector of TGNap1$^{1-300}$ was generated by PCR and cloned into pCR8 vector. The resulting entry vector was then cloned into pEarleyGate by LR reaction. The plant expression vector was transformed into agrobacteria and verified by sequencing. Stable transgenics were generated by the floral-dipping method. MIN7 was amplified from MIN7-GFP by PCR and cloned into pCR8 vector. The resulting vector was cloned into a modified pEarleyGate102-mCherry by LR reaction. The construct was verified by sequencing and transformed into a plant expression vector.

## Extraction of apoplastic fluid

*Arabidopsis* leaves were infiltrated with either mock or *Pst* DC3000. Leaves were detached 24 hpi and vacuum-infiltrated for 3–5 min with 1× PBS (pH 7.4) containing 0.004% Silwet L-77. The vacuum-infiltrated leaves were pat-dried to remove excess buffer. Leaves were arranged into a 10-ml syringe inserted into a 50-ml conical tubes. Conical tubes were spun at 5000× *g*, 10 min at 4 °C to collect apoplastic fluid. In all, 5 μL of the apoplastic fluid was used to determine protein concentration and appropriate amount of loading buffer was added. Samples were subjected to SDS-PAGE and immunoblot analyses using anti-PR1 (1:2500, Agrisera), anti-PR2 (1:1000, Agrisera) and anti-cFPbase (1:5000, Agrisera).

For oryzalin treatments, *Arabidopsis* leaves were infiltrated with either mock + 100 μM oryzalin or *Pst* DC3000 + 100 μM oryzalin.

## Proteomics

**Proteolytic digestion.** Protein pellets were resuspended in 270 μL of 100 mM Tris-HCl (pH 8.5) supplemented to 4% (w/v) sodium deoxycholate (SDC). Samples were reduced and alkylated by adding TCEP and chloroacetamide at 10 mM and 40 mM, respectively and incubated for 5 min at 45 °C with shaking at 2000 rpm in an Eppendorf Thermo-Mixer C. Trypsin, in 50 mM ammonium bicarbonate, was added at a 1:100 ratio (wt/wt) and the mixture was incubated at 37 °C overnight with shaking at 1500 rpm in the Thermomixer. The final volume of each digest was ~300 μL. After digestion, SDC was removed by phase extraction. The samples were acidified to 1% TFA and subjected to C18 solid phase clean up using StageTips[1] to remove salts[69]. Column eluates were dried by vacuum centrifugation and frozen at −20 °C.

**Isobaric peptide labeling.** Peptide samples were then resuspended in 100 μL of 100 mM triethylammonium bicarbonate (TEAB) and labeled with TMT reagents from ThermoScientific according to the manufacturer's instructions. In total, 2 μL aliquots were taken from each labeled sample and reserved for testing labeling/mixing efficiency by MS[70]. The remaining labeled peptides were mixed 1:1 and purified by solid phase extraction using c18 SepPaks. Eluted peptides dried by vacuum centrifugation to ~2 μL and stored at −20 °C. Purified peptides

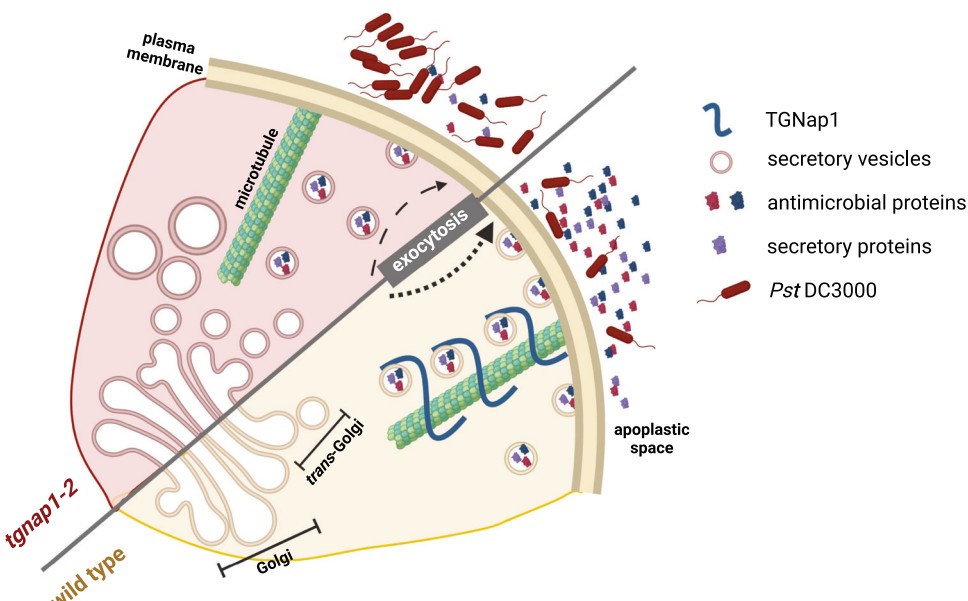

**Fig. 7 | A model for the role of TGNap1 in immunity.** TGNap1 is a microtubule-binding protein and associated with a subpopulation of TGNs, where it impacts TGN budding (Renna et al.[15]). The TGNap1 loss-of-function mutant, *tgnap1-2*, affects endocytic and exocytic traffic in pathogen-unchallenged plants. Upon *Pst* DC3000 infection, secretion of antimicrobial proteins to the apoplast is significantly reduced in *tgnap1-2* leading to increased bacterial growth. The presence of antimicrobial proteins, upon *Pst* DC3000 treatment, in *tgnap1-2* apoplast indicates the existence of alternative pathways employed by plants for apoplastic protein secretion. Oryzalin-mediated disruption of microtubule traffic suggests that proteins other than TGNap1 also need microtubules for secretion to the apoplast.

were resuspended in 2% acetonitrile/0.1%TFA to 20 μL. Labeling efficiency for each set was determined to be: Mock- Col-3/*tgnap1-2* – 94% and *Pst* DC3000 Col-3/*tgnap1-2* – 100%.

**LC/MS/MS analysis.** An injection of 2 μL was automatically made using a Thermo EASYnLC 1200 onto a Thermo Acclaim PepMap RSLC 0.1 mm × 20 mm C18 trapping column and washed for ~5 min with buffer A. Bound peptides were then eluted over 125 min onto a Thermo Acclaim PepMap RSLC 0.075 mm × 500 mm resolving column with a gradient of 5%B at 8%B at 2 min, 8% to 40% at 110 min, 40% to 90% at 115 min and held at 90%B for the duration of the run (Buffer A = 99.9% Water/0.1% Formic Acid, Buffer B = 80% Acetonitrile/0.1% Formic Acid/19.9% Water) at a constant flow rate of 300 nl/min. Column temperature was maintained at a constant temperature of 50 °C using an integrated column oven (PRSO-V2, Sonation GmbH, Biberach, Germany). Eluted peptides were sprayed into a ThermoScientific Q-Exactive HF-X mass spectrometer using a FlexSpray spray ion source. Survey scans were taken in the Orbi trap (120,000 resolution, determined at *m/z* 200) and the top 15 ions in each survey scan are then subjected to automatic higher energy collision-induced dissociation (HCD) with fragment spectra acquired at 45,000 resolution. The resulting MS/MS spectra are converted to peak lists using MaxQuant[71], v1.6.0.1, and searched against a custom database containing all *A. thaliana* protein sequences available from The Arabidopsis Information Network (www.tair.org, v10) and appended with common laboratory contaminants (downloaded from www.thegpm.org, cRAP project) using the Andromeda[72] search algorithm, a part of the MaxQuant environment. The MaxQuant output was then analyzed using Scaffold, v5.0.1 to probabilistically validate protein identifications. Assignments validated using the Scaffold 1%FDR confidence filter are considered true. Andromeda parameters for all databases were as follows: allow up to two missed tryptic sites; Fixed modification of Carbamidomethyl Cysteine; variable modification of Oxidation of Methionine, Deamidation of protein N-term; peptide tolerance of +/− 20 ppm; MS/MS tolerance of +/− 20 ppm; FDR calculated using randomized database search.

**Tobacco transient transformation**
Transient expression of SYP61-CFP, TGNap1-YFP, and MIN7-mCherry was performed using 4-week-old *N. tabacum* (cv Petit Havana) plants by infiltrating a 1:1:1 mixture of Agrobacterium tumefaciens containing each construct (GV3101; $OD_{600} = 0.01$) as described previously[73].

**Confocal laser scanning microscopy**
72 hpi confocal laser scanning microscopy was performed using non-saturating laser levels of pixel fluorescence intensity with a confocal microscope Nikon A1RSi. Three types of fluorescent signals (CFP, ex 443 nm/em 468–503 nm; YFP, ex 514 nm/em 521–554 nm; mCherry, ex 560 nm/em 580–630 nm) were acquired simultaneously. The linear LUT function of the Nikon software (Nis-Elements AR) was applied to enhance the contrast of the image without manipulation (https://www.gvsu.edu/cms4/asset/8FCAC028−902A-3EFC-5137403A360C8843/user_guide_nis-elements_ar.pdf). The quantification analyses of pixel intensity were performed using the raw data with the functions embedded in the Nis-Elements AR. At least 30 distinct puncta per image beyond the background fluorescence level were selected with the ROI detection tool and used for the calculation of Pearson's correlation by the Nis-Elements AR. A total of 1220 punctate structures from 30 images were used for the analyses.

**Graphics**
The model depicted in Fig. 7 was generated using BioRender and assembled using Adobe Illustrator.

**Description of boxplot limits**
Boxplot limits at minima−first quartile (25%), maxima−third quartile (75%), centre−median, whiskers extend to the minimum and maximum values but not further than 1.5 inter-quartile range from respective minima or maxima of the boxplot. Outliers are indicated with *.

**Statistics and reproducibility**
All experiments were designed with requisite positive and negative controls. All experiments were replicated at least three times, unless

otherwise stated. No data were excluded from the analyses, and experiments were not randomized or blinded.

## Reporting summary

Further information on research design is available in the Nature Portfolio Reporting Summary linked to this article.

## Data availability

The RNA-seq data are deposited in the National Center for Biotechnology Information BioProject database with the submission ID–PRJNA922131, which can be accessed using the following link (https://www.ncbi.nlm.nih.gov/bioproject/PRJNA922131). The apoplastic proteomics data are deposited with the MassIVE proteomics database with the submission ID - MSV000092473 and can be accessed using the following link (https://massive.ucsd.edu/ProteoSAFe/dataset.jsp?task=877ea08b93c9429fa94d5f2575f83395). Source data are provided with this paper.

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

## Acknowledgements

We thank colleagues in the Brandizzi lab for helpful discussions. We thank Douglas Whitten at the MSU Proteomics Core Facility and Tony Schilmiller at MSU Mass Spectrometry and Metabolomics Core Facility for technical guidance with the proteomics and metabolomic analyses. We thank Syngenta Crop Protection, LLC for providing BTH. This work was funded primarily by the MSU Research Foundation (F.B.). We also acknowledge partial support from the US Department of Energy Great Lakes Bioenergy Research Center (DOE BER Office of Science DE-SC0018409), US Department of Energy, under FWP# 100878 program at SLAC National Accelerator Laboratory, under contract DE-AC02-76SF00515, the Chemical Sciences, Geosciences and Biosciences Division, Office of Basic Energy Sciences, Office of Science, US Department of Energy (award number DE-FG02-91ER20021) and National Institutes of Health (R35GM136637) to F.B., and National Institutes of Health (1R01AI155441) to S.Y.H.

## Author contributions

D.D.B. and F.B. conceived and designed the study; D.D.B., S.J.K., and K.N. performed experiments; D.D.B., D.K.K., S.Y.H., and F.B. analyzed and interpreted the data, and D.D.B. and F.B. wrote the manuscript.

## Competing interests

The authors declare no competing interests.
