## [Peer Review File · Nature Communications]

Defense against phytopathogens relies on efficient antimicrobial protein secretion mediated by the microtubule-binding protein TGNap1REVIEWER COMMENTS

Reviewer #1 (Remarks to the Author):

Secretion of antimicrobial compounds to the extracellular space is one of the general strategies plants use to defend themselves against pathogens. This defense related secretion is thought to involve post-Golgi trafficking as well as coordination of the cortical cytoskeleton, however, the detailed cellular pathways and molecular players remain largely unknown. In this manuscript, by studying the plant immunity against the pathogenic bacterium *Pseudomonas syringae* pv. tomato DC3000, Bhandari et al. discovered a new pathway for defense-related secretion which is mediated by the previously characterized TGN-associated and microtubule (MT)-binding protein TGNap1. The authors also demonstrated that this MT-related TGN/TGNap1 pathway does not overlap with other known TGN-associated pathways including the SA and MIN7 pathways. This finding adds a new mechanism that advances our understanding of cortical MT-mediated post-Golgi trafficking in plant immunity and should be of interest to researchers from a broad field.

The manuscript is well written and organized; the experiments are generally well designed, controlled and analyzed. However, I have a few suggestions for the authors to further improve this work, and there are a few minor points that need clarification:

Major points:

1. The protein immunoblot assay in Fig. 4c is a key experiment to support the main role of TGNap1 in defense related secretion. The authors should add the TGNap1-YFP complementation line as a necessary control to rule out the possibility that the reduction of PR1 is due to other independent effects. Moreover, there seems to be only one repeat of this experiment (no error bars on the protein quantification plot). At least 3 repeats should be done to be able to compare the protein levels statistically (same problem for Fig. 5a).
2. The proteomics assay in *tgnap1* mutant (Fig. 4c) identified 18 proteins that had increased abundance in the apoplastic fluid; some even had higher fold changes than the reduced ones. Would this suggest that the role of TGNap1 is disrupting the homeostasis of trafficking, and perhaps perturbing endocytosis, rather than only on the secretion side? And is it possible that the increased secretion of proteins also contributes to the overall increased susceptibility of this mutant? Authors should discuss these possibilities in the Discussion.
3. TGNap1 also plays a role in endocytosis as described in the group's previous paper (Renna et al., 2018). Did the authors look into whether TGNap1 affects endocytosis during immunity, e.g. by using the well characterized ligand-induced receptor mediated endocytosis of FLS2 during PTI? Since the authors showed that TGNap1 at least partially contributes to flg22-induced PTI (Fig. 2d), they are encouraged to dig deeper into TGNap1's role during PTI and test whether it is due to affecting the endocytic trafficking of FLS2 or altering early hallmark events associated with signaling cascades. Experiments like measuring the apoplastic ROS burst, calcium influx, or MAPK phosphorylation following flg22 elicitation should be feasible.
4. The authors prepared complementation lines with both full length and truncated TGNap1 (MT-binding domain only) tagged with YFP; with such genetic materials in hand, they should be able to examine at the cellular level the dynamic localization patterns of the tagged TGNap1 versus the truncated version during Pst infection. This may help elucidate the mode of action of this TGN population at the subcellular level, and since the authors claimed that the function of TGNap1 is only partially dependent on MTs, such experiments may help further clarify this point.

Minor points:

1. Immunoblots in Fig. 1d, 4c and 5a: no protein ladders or the size of the protein ladders are

not labeled on the gels. No loading control for Fig. 4c. No description in the Methods on how the protein levels are quantified in immunoblot assays.

2. Please define what L2FC is in Fig. 3a legend.

3. In Line 141, not sure that the description “extracellular defense” is accurate here, since FLS2 mediated defense also occur intracellularly.

4. Line 312-313, “TGNap1 operates in a defense pathway that only partially overlaps with the immune pathway dependent on TGN-localized MIN7”; however, authors indicate in the Results section that TGNap1 and MIN7 are involved in separate pathways and do not overlap.

5. In line 83, 198 and a few other places, the citation “Renna et al., 2018” need to be changed to numbers.

Reviewer #2 (Remarks to the Author):

In this newly submitted Ms, Bhandari and co-authors discovered that the novel TGN-localized protein TGNap1, which is identified by the author’s group using microscopy-based forward genetic screen on EMS mutagenized *Arabidopsis thaliana* cotyledon, is essential for the MT-mediated secretion of antimicrobial proteins and plant immunity.

Interestingly, the function of TGNap1 in plant immunity is distinct from the previous identified TGN protein related with plant immune including MIN7 and ECHIDNA. The findings strengthened the concept that plant TGN populations are heterogeneous and functional diverse. In general, the findings are novel and could be important for the fields, not only for the plant immune but also plant cell biology/membrane trafficking. The results are solid and the conclusions are concise. I enjoyed reading the Ms and have several comments/suggestions to further improve the Ms:

Major Concerns:

1. Although TGNap1/MIN7/ECHIDNA are all TGN-localized proteins, the TGNap1 have non-overlapping function with MIN7 and ECHIDNA in plant immunity. Did the TGNap1 exhibited distinct localisation from MIN7/ECHIDNA in plant cell, especially when against pathogen infection?

2. TGNap1 was shown to be essential for the secretion of antimicrobial proteins upon pathogen infection. Are the cargoes including PR1, BBE4, SAP specific for TGNap1 only? How about the MIN7 and ECHIDNA, are they essential for the secretion of these antimicrobial proteins?

3. It would be good if confocal analysis could be performed on the localisation of PR1 and other identified antimicrobial proteins in the *tgnap1-2* upon Pst DC3000 infection. These proteins could be trapped in these subpopulations of TGN essential for the TGNap1-dependent immunity in *tgnap1-2* mutant.

4. To ensure the reliability for the TMT-based proteomic analysis for secreted cargo identification from isolated apoplastic fluids, I would suggested the authors to repeat the TMT analysis for three times for the quantification analysis or at least label the samples again in a reverse order (the label-tag) to confirm the obtained results. (Li et al., *Nature Plants*, 2021) This could preclude the bias of the label-tag used for the assay.

5. For all the immunoblot analysis on the PR1 secretion, please also include an internal control (cFBPase or other cytosolic proteins which are not secreted) to make sure the collected leaf apoplastic fluid were not contaminated by the plant cells.
6. I would recommend to put the results on the gene expression in tgnap1-2 upon Pst DC3000 infection (Figure 3) in supplementary information, since these results may not be an important point for the major conclusion in the current Ms.
7. Will the TGNap1 tightly associate (interact and colocalize) with MTs upon pathogen infection?
8. As mentioned by the authors, the first 182 aa of TGNap1 are sufficient to bind MTs as shown in their previous study. Nonetheless, the 1-300aa TGNap1 was used for the complementation assay. I would suggest to use the 182 aa construct for the complementation.
9. It would be good if Taxol could be included for the analysis on the relationship between TGNap1 and MTs and the functions of TGNap1 in immunity. [L¹SEP]
10. Any indications or hints on how the antimicrobial proteins are sorted in TGN and transported to PM? This is an interesting/unknown question in plant cell biology field. The authors could provide some discussions on these.

Minor comments:

1. Line 83, Page 5. The format of the citation (Renna et al., 2018) should be number as in the whole Ms.
2. Line 148, Page 7. Untreated and mock-treated plants should be elaborated.
3. Line 175, Page 7. It would be good to name out the three genes which were up-regulated in Col-3 but down-regulated in tgnap1-2, and the four genes that were down-regulated in Col-3 but up-regulated in tgnap1-2 in the text. [L¹SEP]
4. Please mark the band size for all the immunoblot results.
5. A model could be included to summarise the findings.

Reviewer #3 (Remarks to the Author):

The manuscript by Bhandari et al., describes the characterization of TGNap-1 in basal immunity. With an elegant series of experiments the authors show that lines mutated in TGNap-1 are susceptible to virulent Pst DC3000 but this is not due to compromised PTI or ETI or Salicylic acid immune signaling. Transcriptional profiling supports these results but PR1 accumulation in the apoplast is compromised. Proteomics on the apoplastic fluid shows compromised apoplastic protein accumulation of mainly defence related proteins, suggesting a selective exocytosis mechanism is affected. Since TGNap-1 was previously shown to associate with microtubules, wildtype and TGNap-1 mutants are challenge with microtubule disrupting drug oryzalin, which compromises wildtype immunity to Pst Dc3000 but does not further compromise tgnap1 mutant resistance. Since partial complementation can be achieved with a TGNap-1 truncated version that maintains the microtubule binding domain, this suggests that TGNap-1 microtubule binding is required but not sufficient. Finally, the authors investigate whether TGNap-1 and TGN associated protein MIN7 act in the same

pathway. Since a double *tnap-1 min7* mutant shows enhanced susceptibility as compared to TGNap-1 and *min7* single mutant the authors conclude that TGNap-1 and MIN7 act separately TGN based immune response.

The results are clearly presented and logically ordered and support the main conclusions that TGNap-1 acts in exocytic trafficking of proteins required for anti-bacterial immunity.

I have just a few comments.

While the data convincingly shows that TGNap-1 is required for basal immunity to bacterial pathogen Pst DC3000, but to what extent is this specific to bacterial pathogens? Have the authors checked whether other bacterial pathogen and fungal/oomycete pathogen infections are also enhanced in the *tnap-1* mutant.

While *flg22* treatment can induced pamp-triggered immunity in the *tnap-1* as show by in fig 2d, suggesting that PTI responses are not affected, one can not exclude that PTI signalling might be affected and contribute to the enhanced susceptibility of the *tnap-1*. To more conclusively show that PTI is not required the authors could test PAMP-triggered ROS production and or MAPK activation.

Partial complementation of the *tnap-1* mutant with the a C-terminal truncated TGNap-1:YFP fusion containing a microtubule binding domain suggest that microtubule binding by TGNap-1 is required, but the obvious loss-of-function experiment, using TGNap-1 lacking the MT-binding domain or mutated in this domain would make the argument stronger

Minor comments

The description of the LC-MS/MS analysis in the method section suggests the data is analysis using MaxQuant, but the data uploaded into the Scaffold software is described to come from a different search engine (Mascot). Could the authors clarify the data analysis in particular which search engine was used?

The proteomics and transcriptomics data sets should be made available through deposition at the appropriate repositories.

Fig 1 D is missing a size of marker indication as well as indication of the expected size TGNap-1-YFP ?

Fig 6b Pst deltaEM needs to be defined in the figure legend

REVIEWER COMMENTS

We thank the reviewers for their supportive and constructive comments. Addressing them allowed us to improve the quality of the work. As the reviewers will see in the responses below, we have endeavored to address the points by adding new experiments and editing the manuscript text and figure where necessary. We hope that the reviewers will find our efforts satisfactory.

Reviewer #1 (Remarks to the Author):

Secretion of antimicrobial compounds to the extracellular space is one of the general strategies plants use to defend themselves against pathogens. This defense related secretion is thought to involve post-Golgi trafficking as well as coordination of the cortical cytoskeleton, however, the detailed cellular pathways and molecular players remain largely unknown. In this manuscript, by studying the plant immunity against the pathogenic bacterium *Pseudomonas syringae* pv. tomato DC3000, Bhandari et al. discovered a new pathway for defense-related secretion which is mediated by the previously characterized TGN-associated and microtubule (MT)-binding protein TGNap1. The authors also demonstrated that this MT-related TGN/TGNap1 pathway does not overlap with other known TGN-associated pathways including the SA and MIN7 pathways. This finding adds a new mechanism that advances our understanding of cortical MT-mediated post-Golgi trafficking in plant immunity and should be of interest to researchers from a broad field.

The manuscript is well written and organized; the experiments are generally well designed, controlled and analyzed. However, I have a few suggestions for the authors to further improve this work, and there are a few minor points that need clarification:

Major points:

1. The protein immunoblot assay in Fig. 4c is a key experiment to support the main role of TGNap1 in defense related secretion. The authors should add the TGNap1-YFP complementation line as a necessary control to rule out the possibility that the reduction of PR1 is due to other independent effects. Moreover, there seems to be only one repeat of this experiment (no error bars on the protein quantification plot). At least 3 repeats should be done to be able to compare the protein levels statistically (same problem for Fig. 5a).

We thank the reviewer for the suggestion of adding TGNap1-YFP complementation line. In the revised manuscript we have performed the suggested experiments and provided the new data set in Fig. 4b. In addition, based on the suggestions from the other reviewers, we have added new protein blots probed with PR2 antibody and a cytosolic marker (cFBPase). The earlier Fig.4 has now been moved to Fig. S5. All the experiments presented in the current manuscript represent consensus data from a minimum of three independent experiments. We elected to normalize band intensities within a blot as this avoids exposure differences that can be misleading. Accordingly, all the blots presented in the manuscript are normalized to the wild type. Hence, the protein quantification plots do not have an error bar. We also provide independent blots in new Fig. S6, which are apoplastic extracts from an independent experiment. As it can be observed from multiple blots with multiple probes and in the unbiased

proteomics datasets, the tgnap1-2 mutant is deficient in the secretion of immune proteins, which, as the reviewer rightly points, out is the key conclusion.

2. The proteomics assay in tgnap1 mutant (Fig. 4c) identified 18 proteins that had increased abundance in the apoplastic fluid; some even had higher fold changes than the reduced ones. Would this suggest that the role of TGNap1 is disrupting the homeostasis of trafficking, and perhaps perturbing endocytosis, rather than only on the secretion side? And is it possible that the increased secretion of proteins also contributes to the overall increased susceptibility of this mutant? Authors should discuss these possibilities in the Discussion.

Thank you for the suggestion. In the revised text we have discussed the possibility of disrupted endocytosis of these proteins having a role in the susceptibility of tgnap1-2 (lines 394-399). To summarize, the flg22-protection assays (Fig. 2d) and MPK phosphorylation data (newly added. Fig. S1c) argue for a more prominent role of disrupted exocytosis. It could very well be possible that, against a pathogen that targets or depends on the endocytosis of the 18 proteins with increased abundance in tgnap1-2, the resistance response could be different..

3. TGNap1 also plays a role in endocytosis as described in the group's previous paper (Renna et al., 2018). Did the authors look into whether TGNap1 affects endocytosis during immunity, e.g. by using the well characterized ligand-induced receptor mediated endocytosis of FLS2 during PTI? Since the authors showed that TGNap1 at least partially contributes to flg22-induced PTI (Fig. 2d), they are encouraged to dig deeper into TGNap1's role during PTI and test whether it is due to affecting the endocytic trafficking of FLS2 or altering early hallmark events associated with signaling cascades. Experiments like measuring the apoplastic ROS burst, calcium influx, or MAPK phosphorylation following flg22 elicitation should be feasible.

We thank the reviewer for the point raised about the role of TGNap1 in endocytosis because addressing it helped us strengthen our earlier results. Indeed, as the reviewer correctly mentions TGNap1 has a role in both endocytosis and exocytosis. Following the suggestion from this reviewer, we have now monitored MPK phosphorylation in tgnap1-2 upon flg22 treatment (Fig. S1c). The results from this experiment are described in lines 102-108. Briefly, we observe that, in line with the flg22 protection assay (Fig. 2d), MPK3/6 phosphorylation is comparable between Col, tgnap1-2 and TGNap1-YFP, reinforcing a limited role of TGNap1 in immunity activated endocytosis and that the hyper-susceptibility of tgnap1-2 can largely be attributed to defective secretion.

4. The authors prepared complementation lines with both full length and truncated TGNap1 (MT-binding domain only) tagged with YFP; with such genetic materials in hand, they should be able to examine at the cellular level the dynamic localization patterns of the tagged TGNap1 versus the truncated version during Pst infection. This may help elucidate the mode of action of this TGN population at the subcellular level, and since the authors claimed that the function of TGNap1 is only partially dependent on MTs, such experiments may help further clarify this point.

We agree with the reviewer that it would be interesting to understand the dynamics of TGNap1 during Pst infection. However, at the four-week-old development stage when Pst assays are generally performed, it is very hard to discern the signal of TGNap1 from the background in live-cell imaging (even in untreated and mock-treated plants). As evidenced by our Western blot results, at this developmental stage, TGNap1 is expressed but its signal is weak, in accordance with the live-cell imaging analyses. Therefore, although the results underscore the occurrence of an interesting regulation of the TGNap1 protein levels, the experiment is unfortunately not feasible.

Minor points:

1. Immunoblots in Fig. 1d, 4c and 5a: no protein ladders or the size of the protein ladders are not labeled on the gels. No loading control for Fig. 4c. No description in the Methods on how the protein levels are quantified in immunoblot assays.

Thank you. All the protein band sizes have been labelled and an explanation of protein quantification has been added to the methods.

2. Please define what L2FC is in Fig. 3a legend.

Thank you, we have abbreviated L2FC – Log 2 fold change.

3. In Line 141, not sure that the description “extracellular defense” is accurate here, since FLS2 mediated defense also occur intracellularly.

This line has been modified in the new text alongside the addition of MPK phosphorylation data.

4. Line 312-313, “TGNap1 operates in a defense pathway that only partially overlaps with the immune pathway dependent on TGN-localized MIN7”; however, authors indicate in the Results section that TGNap1 and MIN7 are involved in separate pathways and do not overlap.

With the addition of the new data on TGNap1 and MIN7 co-localization, we have amended the relevant text sections to highlight the functional relationship between the two proteins.

5. In line 83, 198 and a few other places, the citation “Renna et al., 2018” need to be changed to numbers.

Thank you for pointing out, we apologize for the reference manager error, which we have rectified in the revision.

Reviewer #2:

In this newly submitted Ms, Bhandari and co-authors discovered that the novel TGN-localized

protein TGNap1, which is identified by the author's group using microscopy-based forward genetic screen on EMS mutagenized *Arabidopsis thaliana* cotyledon, is essential for the MT-mediated secretion of antimicrobial proteins and plant immunity.

Interestingly, the function of TGNap1 in plant immunity is distinct from the previous identified TGN protein related with plant immune including MIN7 and ECHIDNA. The findings strengthened the concept that plant TGN populations are heterogeneous and functional diverse. In general, the findings are novel and could be important for the fields, not only for the plant immune but also plant cell biology/membrane trafficking. The results are solid and the conclusions are concise. I enjoyed reading the Ms and have several comments/suggestions to further improve the Ms:

Major Concerns:

1. Although TGNap1/MIN7/ECHIDNA are all TGN-localized proteins, the TGNap1 have non-overlapping function with MIN7 and ECHIDNA in plant immunity. Did the TGNap1 exhibited distinct localisation from MIN7/ECHIDNA in plant cell, especially when against pathogen infection?

*Thank you for the suggestion. We have now added new data in Fig. 6a to test the localization of MIN7 and TGNap1 to the TGN. Our data point to a partial co-localization between the pair of TGNap1/MIN7 (63.9%). Combined with the increased susceptibility of the *tnap1-2 min7* double mutant, these results (now discussed in lines 312-324 and 433-436xxx-xxx and xxx-xxx) point to a functional complexity of TGN populations.*

*Unlike *tnap1-2* and *min7*, the *echidna* mutant leads to hyper activation of SA and SA-dependent resistance (Liu et al., 2023, Plant Physiology), but the role of TGNap1 in immunity is independent of SA (Fig. 2). While a functional relationship between *Echidna* and TGNap1 remains unknown and beyond the scope of this manuscript, based on Liu et al. and our data, it can be inferred that the two proteins have differing roles in immunity.*

2. TGNap1 was shown to be essential for the secretion of antimicrobial proteins upon pathogen infection. Are the cargoes including PR1, BBE4, SAP specific for TGNap1 only? How about the MIN7 and ECHIDNA, are they essential for the secretion of these antimicrobial proteins?

*In *tnap1-2*, the evidence that the cargoes mentioned in the manuscript, including PR1, PR2, BBes, are reduced in the apoplast suggests the existence of other trafficking pathways working redundantly with TGNap1. The role of *Echidna* or MIN7 in the secretion of these proteins is beyond the scope of this manuscript. Interestingly, Liu et al. 2023 also showed a decrease in apoplastic accumulation of antimicrobial proteins such as PR1 and PDF1 in the *echidna* mutant. We have referred the readers to these findings in the revised manuscript.*

3. It would be good if confocal analysis could be performed on the localisation of PR1 and other identified antimicrobial proteins in the *tnap1-2* upon Pst DC3000 infection. These proteins could be trapped in these subpopulations of TGN essential for the TGNap1-dependent immunity in *tnap1-2* mutant.

Thank you. We agree with the reviewer, but instead of using confocal imaging, we have resolved with using quantitative proteomics analyses of the apoplastic space. This approach allowed us to infer quantitatively that the antimicrobial proteins are not efficiently secreted. This

is in line with our previous manuscript (Renna et al., 2018, Nature communications) where we reported that the loss of TGNap1 leads to a reduced exocytosis. Quantitative proteomics measure quantifiable changes in the apoplastic protein content. Such an unbiased approach allowed us to analyze quantitatively the efficiency of trafficking for selected cargo broadly rather than focusing on selected antimicrobial cargo.

4. To ensure the reliability for the TMT-based proteomic analysis for secreted cargo identification from isolated apoplastic fluids, I would suggested the authors to repeat the TMT analysis for three times for the quantification analysis or at least label the samples again in a reverse order (the label-tag) to confirm the obtained results. (Li et al., Nature Plants, 2021) This could preclude the bias of the label-tag used for the assay.

Thank you for the suggestion and we appreciate the reviewer's concern. The TMT-based approach is the most reliable for unbiased proteomics. There are multiple published studies (Park et al., Nature Communications, 2022; Gabaev et al., Cell reports, 2020; Udeshi et al., Nature Communications, 2020), which have performed one TMT experiment with variable "n" for each sample group tested. Please also note that the apoplastic fluid collected for the proteomics experiment was obtained from three to four independent experiments which would account for any biological variation. In addition, the high labelling efficiency (94-100%) would preclude any plausible tagging bias.

5. For all the immuoblot analysis on the PR1 secretion, please also include an internal control (cFBPase or other cytosolic proteins which are not secreted) to make sure the collected leaf apoplastic fluid were not contaminated by the plant cells.

Thank you for the suggestion. We have now added cFBPase as a control to determine the level of cytosolic overflow (which is not completely avoidable) during isolation of apoplastic fluid.

6. I would recommend to put the results on the gene expression in tgnap1-2 upon Pst DC3000 infection (Figure 3) in supplementary information, since these results may not be an important point for the major conclusion in the current Ms.

Thank you for the suggestion. We would like to retain the tgnap1-2 transcriptome data in fig. 3 as we believe that this dataset is crucial in establishing that the transcriptome of wild type and tgnap1-2 mutant upon Pst DC3000 infection is comparable and that the susceptibility of tgnap1-2 lies in trafficking of immune-related proteins and not in their timely production.

7. Will the TGNap1 tightly associate (interact and colocalize) with MTs upon pathogen infection?

Thank you for the interesting query. We have attempted to study this. However, it is difficult to reliably visualize TGNap1 in plants at four-week-old stage when it is amenable to perform the pathogen assays. As also evidenced by our Western blot results, at this developmental stage, TGNap1 is expressed, but its signal is weak, in accordance with the live-cell imaging analyses. If we were to speculate, we would posit that TGNap1 interaction with MTs will not change upon

pathogen infection, as removal of TGNap1 and co-treating with oryzalin had no additional susceptibility, suggesting that the pathways converge.

8. As mentioned by the authors, the first 182 aa of TGNap1 are sufficient to bind MTs as shown in their previous study. Nonetheless, the 1-300aa TGNap1 was used for the complementation assay. I would suggest to use the 182 aa construct for the complementation.

Thank you. Indeed, the first 182 aa of TGNap1 were sufficient to bind MTs in vitro (Renna et al., 2018); however in analyzing the predicted protein secondary structure of TGNap1, we noted the existence of two putative coiled-coil domains (215-249 and 257-291 a.a). Therefore, in generating a truncated version of TGNap1 we included this region also. Our hypothesis was that these coiled-coil domains are important in providing structural integrity or for binding to vesicles. It is important to note that the reciprocal of this construct (TGNap1²⁹⁷⁻¹¹⁸²) was unstable and therefore could not be used for analyses. Please also see our response to Reviewer#3 point #3.

9. It would be good if Taxol could be included for the analysis on the relationship between TGNap1 and MTs and the functions of TGNap1 in immunity.

Thank you for the suggestion. We have now added new data on the effect of taxol during Pst infection in Fig. S7 and discussed the results in lines 286-290.

10. Any indications or hints on how the antimicrobial proteins are sorted in TGN and transported to PM? This is an interesting/unknown question in plant cell biology field. The authors could provide some discussions on these.

We have refrained from speculating on the mechanisms for cargo sorting at the TGN and the role of TGNap1 on this trafficking aspect. As the reviewer rightly points out, there is a paucity of research in this area, perhaps due to redundancy of the pathways involved. We are concerned that a discussion on this topic may be a distraction, but we agree with the reviewer that this is a very interesting question in plant biology (and cell biology at large).

Minor comments:

1. Line 83, Page 5. The format of the citation (Renna et al., 2018) should be number as in the whole Ms.

Thank you for pointing out the reference manager error, we have reformatted references to the Nature communications format.

2. Line 148, Page 7. Untreated and mock-treated plants should be elaborated.

We have elaborated the difference between untreated (no pre-treatment) vs mock (DMSO) treatment in the revised text.

3. Line 175, Page 7. It would be good to name out the three genes which were up-regulated in

Col-3 but down-regulated in *tgnap1-2*, and the four genes that were down-regulated in Col-3 but up-regulated in *tgnap1-2* in the text.

Thank you, these genes have been called out in the revised text.

4. Please mark the band size for all the immunoblot results.

Thank you, we have labelled band sizes in immunoblots.

5. A model could be included to summarise the findings.

We have added a model summarizing our findings in new Fig. 7

Reviewer #3 (Remarks to the Author):

The manuscript by Bhandari et al., describes the characterization of TGNap-1 in basal immunity. With an elegant series of experiments the authors show that lines mutated in TGNap-1 are susceptible to virulent Pst DC3000 but this is not due to compromised PTI or ETI or Salicylic acid immune signaling. Transcriptional profiling supports these results but PR1 accumulation in the apoplast is compromised. Proteomics on the apoplastic fluid shows compromised apoplastic protein accumulation of mainly defence related proteins, suggesting a selective exocytosis mechanism is affected. Since TGNap-1 was previously shown to associate with microtubules, wildtype and TGNap-1 mutants are challenge with microtubule disrupting drug oryzalin, which compromises wildtype immunity to Pst Dc3000 but does not further compromise *tgnap1* mutant resistance. Since partial complementation can be achieved with a TGNap-1 truncated version that maintains the microtubule binding domain, this suggests that TGNap-1 microtubule binding is required but not sufficient. Finally, the authors investigate whether TGNap-1 and TGN associated protein MIN7 act in the same pathway. Since a double *tgnap-1 min7* mutant shows enhanced susceptibility as compared to TGNap-1 and *min7* single mutant the authors conclude that TGNap-1 and MIN7 act separately TGN based immune response.

The results are clearly presented and logically ordered and support the main conclusions that TGNap-1 acts in exocytic trafficking of proteins required for anti-bacterial immunity.

I have just a few comments.

While the data convincingly shows that TGNap-1 is required for basal immunity to bacterial pathogen Pst DC3000, but to what extent is this specific to bacterial pathogens? Have the authors checked whether other bacterial pathogen and fungal/oomycete pathogen infections are also enhanced in the *tgnap-1* mutant.

*Thank you for the suggestion. We have tested the role of TGNap1 in immunity against the Oomycete pathogen Hyaloperonospora arabidopsidis (Hpa) NOCO2. Our results indicate an increased susceptibility in the *tgnap1-2* mutant upon infection with Hpa. The results of this experiment are presented in Fig. S1d and discussed in lines 113-119. These additional data*

further support a role of TGNap1 in immunity.

While flg22 treatment can induced pamp-triggered immunity in the tgnap-1 as show by in fig 2d, suggesting that PTI responses are not affected, one can not exclude that PTI signalling might be affected and contribute to the enhanced susceptibility of the tgnap-1. To more conclusively show that PTI is not required the authors could test PAMP-triggered ROS production and or MAPK activation.

Thank you for the suggestion. In the revised submission, we present data pertaining to MPK phosphorylation in tgnap1-2 upon flg22 treatment (Fig.S1c). The results from this experiment are described in lines 102-108. To summarize, MPK3/6 phosphorylation is comparable between Col, tgnap1-2 and TGNap1-YFP reinforcing a limited role of TGNap1 in immunity activated endocytosis and that the hyper-susceptibility of tgnap1-2 can largely be attributed to defective exocytic secretion.

Please also see our response to reviewer#1, point 3 for the point about MPK phosphorylation upon flg22 treatment.

Partial complementation of the tgnap-1 mutant with the a C-terminal truncated TGNap1:YFP fusion containing a microtubule binding domain suggest that microtubule binding by TGNap-1 is required, but the obvious loss-of-function experiment, using TGNap-1 lacking the MT-binding domain or mutated in this domain would make the argument stronger

Thank you for the suggestion. We had a similar line of investigation when we generated the truncated version of TGNap1. The TGNap1 truncation lacking the MT-binding domain (TGNap1 297-1182) was unstable and did not express in planta, likely due to a stabilizing role of the first 300 amino acids. Therefore, we chose to establish a role of TGNap1 in immunity using the microtubule homeostasis disrupting chemicals oryzalin and taxol.

Minor comments

The description of the LC-MS/MS analysis in the method section suggests the data is analysis using MaxQuant, but the data uploaded into the Scaffold software is described to come from a different search engine (Mascot). Could the authors clarify the data analysis in particular which search engine was used?

Thank you for the query. The proteomics methods have been clarified in the revised manuscript. The mention of Mascot was an oversight, it should have read MaxQuant, it has been corrected in the revised methods section.

The proteomics and transcriptomics data sets should be made available through deposition at the appropriate repositories.

Certainly! The data will be made available on the respective repositories, as requested by the journal.

Fig 1 D is missing a size of marker indication as well as indication of the expected size TGNap-1-YFP ?

Thank you. A size indication has been added to Fig. 1d. Please also see Fig. S3b.

Fig 6b Pst deltaEM needs to be defined in the figure legend

Pst delta EM has been defined in the revised figure legend. Thank you.

REVIEWERS' COMMENTS

Reviewer #1 (Remarks to the Author):

The authors have conducted additional experiments, introduced appropriate controls where necessary, and thoroughly addressed our previous comments.

Reviewer #2 (Remarks to the Author):

The authors have addressed most of my concerns in the revised manuscript. I would be happy to see the Ms published in Nature Communications. Congratulations!

Reviewer #3 (Remarks to the Author):

The authors have adequately addressed the questions and comments I had and I have no further requests for revisions